

# Modeling snow slab avalanches caused by weak layer failure – Part II: Coupled mixed-mode criterion for skier-triggered anticracks

Philipp L. Rosendahl[1,2,*] and Philipp Weißgraeber[1,3,4,*]

[1]2$\varphi$ GbR, www.2phi.de, Tübingen, Germany
[2]Technische Universität Darmstadt, Department of Mechanical Engineering, Germany
[3]Robert Bosch GmbH, Corporate Research and Advance Engineering, Renningen, Germany
[4]ARENA2036 research campus, Universität Stuttgart, Germany
[*]Contributed equally to this work.

**Correspondence:** mail@2phi.de

**Abstract.** Using the analytical model presented in part I of this two-part paper, a new conceptual understanding of anticrack nucleation in weak layers is proposed. To obtain a sufficient condition for onset of failure two necessary conditions must be satisfied simultaneously: i) The weak layer must be overloaded in terms of stress and ii) the initiating crack must release enough energy for the formation of new surfaces. This so-called coupled criterion was proposed by Leguillon (2002) [Eur J Mech-A

Solid, 21(1), 61–72, 2002]. No assumptions on initial defects within the weak layer are needed. Instead, the failure criterion provides both critical loading and the size of initiating cracks. It only requires the fundamental material properties strength and fracture toughness as inputs. Crack initiation and subsequent propagation are covered by a single criterion containing both a strength-of-materials and a fracture mechanics condition.

Analyses of skier-loaded snowpacks show the impact of slab thickness and slope angle on critical loading and crack initiation

length. In the limit cases of very thick slabs and and very steep slopes we obtain natural avalanche release. A discussion of different mixed-mode stress and energy criteria reveals that a wrong choice of mixed-mode hypotheses can yield unphysical results. The effect of material parameters such as density and compliance on weak layer collapse is illustrated.

The framework presented in this two-part series harnesses the efficiency of closed-form solutions to provide fast and physically sound predictions of critical snowpack loads using a new conceptual understanding of mixed-mode weak layer failure. It

emphasized the importance of both stress and energy in avalanche release.

## 1   Introduction

To study the onset of weak layer failure fracture mechanics models have been proposed that extend the classical stability indexes. Classical fracture mechanics is restricted to the analysis of growth of existing cracks. For a crack to propagate, a sufficient energy release is required to overcome the energy barrier for crack growth, that originates, e.g., from surface energy

and dissipative processes. Fracture mechanics has been applied to the analysis of weak layer shear cracks (McClung, 1979; Louchet, 2001; Bažant et al., 2003) and to the propagation of weak layer collapse (Heierli, 2005; Heierli and Zaiser, 2008).





The latter led to the anticrack[1] concept for avalanche release (Heierli and Zaiser, 2008; Heierli et al., 2008). Rendering part of the failure process as a collapse of the weak layer and describing it using fracture mechanics created a new perspective on avalanche release. The concept provides a physical explanation for remote-triggering and whumpf sounds in avalanche-prone terrain.

5    In recent years fracture mechanics approaches were used and improved by many researchers (e.g. Sigrist and Schweizer, 2007; van Herwijnen and Jamieson, 2007; Gauthier and Jamieson, 2008a; Gaume et al., 2015). Their use requires an existing crack or flaw within the loaded structure. Hence, in many cases assumptions on the size of such cracks are made (for instance meso-scale "super-weak zones" in weak layers (Bader and Salm, 1990) or virtual cracks (Waddoups et al., 1971)). Investigating skier-triggered weak layer failure it is often assumed that skier loading causes micro-scale damage leading to localized flaws 10 (Schweizer, 1999; Schweizer et al., 2003). Several studies analyzed the size of such flaws (McClung and Schweizer, 1999; Schweizer, 1999; Gaume et al., 2017). That is, the initiation of local defects is considered to happen in a separate process and maybe even at a different time-scale (McClung, 1981) than the propagation of the fracture itself. Assuming skier loading can create uncritical defects implies that several uncritical loading events in temporal sequence can accumulate to a critical size. In other words, such models would predict avalanche release if only enough skiers ski the same slope in close temporal succession 15 which is an inherent flaw.

    In order to characterize the resistance against crack propagation a new field test, the propagation saw test (PST), was developed. It was first mentioned by van Herwijnen and Jamieson (2005) and described in detail by Gauthier and Jamieson (2008b). The PST is very similar to the double cantilever beam (DCB) test – an established lab test to quantify fracture mechanics material properties. The latter is standardized (ASTM Standard E399-17, 2017; ASTM Standard D3433-99, 2012), used by 20 researchers and in industrial applications and proved to allow for reliable measurements of the fracture toughness (Rosendahl et al., 2019).

    To link stress-based stability indices and fracture mechanics criteria, combined models were proposed. Gaume and Reuter (2017) compare critical crack lengths obtained from a discrete element study to the size of overloaded areas of weak layers to analyze the ability of flaws to initiate and also propagate. The size of the overloaded area is estimated from closed-form ana- 25 lytical solutions for skier-loaded snowpacks (Föhn, 1987; Monti et al., 2015). Their parametric study shows realistic parameter dependencies and includes the case of self-triggered natural avalanche release without additional load. Although the critical crack length considered by Gaume et al. (2017) is not a direct fracture mechanics material property, this work is among the first to directly link stability indices (which base on strength-of-materials) and fracture mechanics. Similar efforts were made by Chiaia et al. (2008) who consider shear failure only and Reuter et al. (2015) who correlate a new stability index based on 30 finite element analyses with a model for critical lengths for mixed-mode crack propagation given by Heierli (2008).

---

[1]Here, anticrack refers to the extension of a collapse (Fletcher and Pollard, 1981). This allows for direct use of the concepts of classical fracture mechanics with the only difference being the sign of the crack tip displacement field. The local displacement field exhibits the same behavior in compression as in tension. It is proportional to the square root of the distance to the crack tip. This should not be confused with the case of a rigid line inclusion, which is also termed "anticrack" which coincidentally shows the same stress singularity (Dundurs and Markenscoff, 1989).



Besides mechanical models which aim for closed-form analytical descriptions of the processes within the snowpack, numerical models were developed to study nonlinear failure processes within the snowpack. Mahajan et al. (2010) provide a comprehensive numerical model using a cohesive zone approach for weak layer fracture. Using this established process zone approach they study mixed-mode failure criteria and the competition of shear and collapse. Gaume et al. (2015, 2017) use

the discrete element method to analyze weak layer fracture in propagation saw tests. In order to model macro-scale crack propagation the particle size in the simulations was chosen significantly larger than microstructural lengths. The results are in good agreement with experimentally obtained critical crack lengths and also with deformations derived from particle tracking velocimetry (Bobillier et al., 2018). The very recent model proposed by Gaume et al. (2018) uses the framework of the material point method with a plastic flow rule for hardening which is modified to account for local softening of the weak layer. In their

model, the volumetric plastic strain of the porous weak layer controls the energy dissipation of the fracture process. The model recovers the onset of weak layer failure, the propagation of the failure across a slope as well as the release and downslope flow of the avalanche (Savage and Hutter, 1989; Christen et al., 2010). An overview of numerical modeling efforts is given by Podolskiy et al. (2013). Numerical methods are typically more general, i.e. more competing effects can be studied within the same model. However, due to computational efforts parametric studies are expensive and general parameter dependencies

cannot be derived directly. They can only be derived from observed results of numerical studies and might not apply outside of considered parameter domains.

The present work aims at providing a physical explanation for skier-triggered anticrack nucleation in weak layers. For this purpose we propose a unified failure criterion which directly links strength-of-materials and fracture mechanics. The criterion accounts for mixed-mode shear and compression loading. We employ closed-form analytical expressions for weak

layer stresses and energy release rates of cracks within the weak layer given in part I of this series.

## 2 Finite fracture mechanics criterion for skier-triggered anticrack nucleation

Dry snow slab avalanche release is typically preceded by the nucleation of an anticrack within the weak layer. As discussed by many researchers, skier-triggered weak layer collapse is a problem of both strength-of-materials and fracture mechanics (see e.g. Schweizer et al., 2003; Gaume and Reuter, 2017). In the following, we will show that applying either one or the other

exclusively yields contradictory and nonphysical predictions in certain situations. This is because infinitesimal crack growth is impossible when no initial crack is present and stresses at existing crack tips are infinite. However, assuming the sudden nucleation of a crack of finite size and combining both strength-of-materials and fracture mechanics in one unified failure criterion resolves apparent contradictions. Let us illustrate this with the following examples.

### 2.1 Why both strength and toughness matter

Consider the problem of edge crack nucleation in four-point bending tests. Figure 1 shows the results of 299 such experiments on rectangular homogeneous wooden beams reported by Fonselius (1997). Depicted are maximum local stresses at failure computed from the critical loading at failure. Self-similar specimens of different size but constant length to height ratio are





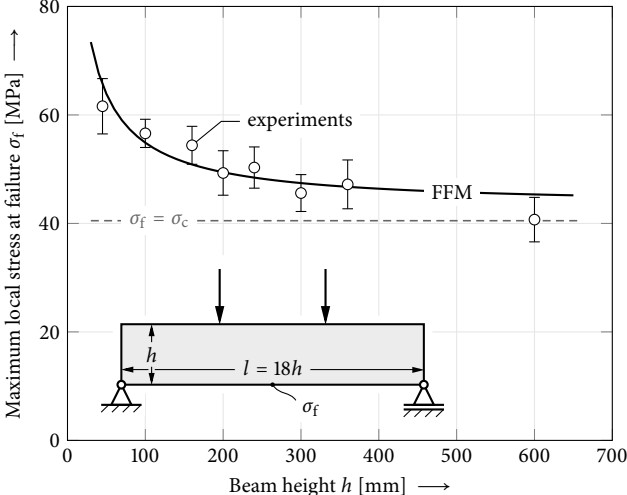

**Figure 1.** Size effect of wood beams in 299 self-similar four-point bending experiments. Experimental results reported by Fonselius (1997) are given as mean and standard deviation. Width of the beam is always $b = 45\,\mathrm{mm}$. Strength-of-materials solution (dashed) and finite fracture mechanics (FFM) prediction (solid) are shown.

tested in this size-effect study. According to strength-of-materials, failure occurs when the maximum stress within the beam reaches the material's strength

$$\sigma_\mathrm{f} = \sigma_\mathrm{c}. \tag{1}$$

However, this simple expression is valid only for sufficiently large beams. The experiments in Figure 1, show that small

5  specimens fail at considerably larger loads than predicted by Eq. (1). The phenomenon is a so-called size effect (Bažant, 1984; Leguillon et al., 2015) and is for instance also present at small holes (Rosendahl et al., 2017) and thin adhesive layers (Stein et al., 2015). In structures with localized stress concentrations the initiation of cracks is observed at loads causing stresses that locally exceed the material's strength. The behavior originates from an insufficient energy release for crack formation.[2] We will come back to this problem in the following showing that modern failure criteria can explain and predict such effects.

10  As a second example, consider a homogeneous isotropic bar subjected to a critical strain $\hat{\varepsilon}$ at which it fails. The critical strain is associated to a critical stress $\hat{\sigma}$. After rupture the broken bar is of course stress and strain-free. It stores no elastic energy. Hence, the energy released during the fracture process is the entire energy which was stored in the bar prior to fracture

$$\Delta\Pi = -\frac{1}{2}Al\hat{\sigma}\hat{\varepsilon} = -\frac{1}{2}Al\frac{\hat{\sigma}^2}{E}, \tag{2}$$

[2]In addition, statistical size effects may play a role as well. As shown by Leguillon et al. (2015), size effects are often a combination of energetic and statistical effects.





where $A$ is the cross section of the bar, $l$ is its length and $E$ its Young's modulus. If the problem is governed by fracture mechanics, the incremental form of the Griffith criterion, $\overline{\mathcal{G}} = -\Delta\Pi/A = \mathcal{G}_\mathrm{c}$, may be used

$$\frac{1}{2}Al\frac{\hat{\sigma}^2}{E} = \mathcal{G}_\mathrm{c}A. \tag{3}$$

It would be wrong to assume that Eq. (3) allows for computing the critical stress at which the bar will break according to

$$\hat{\sigma} = \sqrt{2\frac{E\mathcal{G}_\mathrm{c}}{l}}, \tag{4}$$

if the fracture toughness $\mathcal{G}_\mathrm{c}$ is a known material constant. Eq. (4) implies that the bar will fail at arbitrarily small loads provided it is sufficiently long, which obviously contradicts observations. It draws an incorrect conclusion because the problem is not governed by energy exclusively but certain stress conditions must be satisfied, too.

In order to resolve the contradictions in the above examples, let us reconsider the four-point bending problem. Instead of examining local processes only, i.e. considering local stresses and expecting *infinitesimal* crack growth from the edge, let us assume the sudden formation of a *finite sized* crack. The concept is known as finite fracture mechanics (FFM) and was proposed by Hashin (1996). A finite crack can only nucleate when its energy release is larger than the fracture toughness and when the material's strength is exceeded on the entire potential crack length. This so-called coupled stress and energy failure criterion was proposed by Leguillon (2002).

Let us now consider such finite cracks in thickness direction on the tensile side of the beam. The energy release rate of these cracks increases with their length. Hence, only sufficiently long finite cracks satisfy the energy condition for a given load or, in other words, a given short finite crack only releases enough energy at very high loads. Stresses decrease linearly with distance from the edge. Hence, longer finite cracks require larger external loads to still satisfy the stress condition on the entire finite crack length. If the energy condition requires a certain crack length $\Delta a$, the normalized crack length $\Delta a/h$ is longer for beams of smaller height $h$. Longer normalized crack lengths require higher loads to satisfy the stress condition. Thus, smaller beams can sustain relatively higher external loads and their apparent flexural strength is larger. FFM predicts the effective flexural strength of all 299 four-point bending experiments shown in Figure 1 using only two fundamental material properties: the uniaxial tensile strength and the mode I fracture toughness.

The above examples show that i) fracture processes are governed not by one exclusive but by two conditions simultaneously even if one often hides the other and ii) strength and toughness are independent fundamental material properties and one cannot be computed from the other.

## 2.2 General coupled stress and energy criterion

The concept of finite fracture mechanics is best understood considering the limitations of classical fracture mechanics. The differential energy release rate[3] $\mathcal{G}(a)$ of an infinitesimal crack advance is only nonzero when the crack increment extends an existing crack (crack growth). It increases with the length of the existing crack and is zero when there is no initial crack

---

[3]See section 2.5 in part I for a comprehensive definition of fracture mechanical quantities.





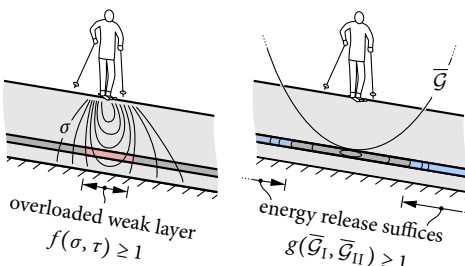

**Figure 2.** Coupled stress and energy criterion for the case of a skier loaded snowpack. On the left the condition of a locally overloaded weak layer and on the right the second condition of sufficient energy release rate are shown. The stress condition has the effect of an upper bound on possible nucleated cracks. Whereas the energy condition provides a lower bound on the crack lengths as short cracks do not release sufficient energy.

($a = 0$) (Weißgraeber et al., 2016). Owing to the vanishing energy release rate of nucleating infinitesimal cracks it must be assumed that the initial crack does not grow infinitesimally from zero length but forms instantaneously and with finite size. The amount of energy $\Delta\Pi$ released per finite crack increment $\Delta a$ is nonzero and known as incremental energy release rate $\overline{\mathcal{G}}(\Delta a) = -\Delta\Pi/\Delta a$.[3] Again, the Griffith criterion must hold and for a finite crack to nucleate the incremental energy release

rate must exceed the fracture toughness:

$$\frac{\overline{\mathcal{G}}(\Delta a)}{\mathcal{G}_{\mathrm{c}}} \geq 1. \tag{5}$$

In case of mixed-mode loading, a generic mixed-mode energy criterion may be formulated according to

$$g\big(\overline{\mathcal{G}}_{\mathrm{I}}(\Delta a),\, \overline{\mathcal{G}}_{\mathrm{II}}(\Delta a),\, \mathcal{G}_{\mathrm{Ic}},\, \mathcal{G}_{\mathrm{IIc}}\big) \geq 1, \tag{6}$$

where $\overline{\mathcal{G}}_{\mathrm{I}}$ and $\overline{\mathcal{G}}_{\mathrm{II}}$ denote incremental energy release rates in mode I and II. Because the incremental energy release rate $\overline{\mathcal{G}}$

increases with increasing size of the finite crack $\Delta a$, a certain minimum finite crack length is required to satisfy Eq. (6). Hence, the energy condition represents a lower bound for the finite crack length $\Delta a$ which is illustrated in the right half of Figure 2.

Besides the critical loading, the size of the initiating finite crack $\Delta a$ is a second unknown. In order to determine both unknowns, a second necessary condition – a stress condition – is required. A crack can only nucleate when the material is overloaded in terms of stress. In case of mixed-mode loading the equivalent stress function $f$ must exceed its critical value in

every point $x$ on the potential surface $\Omega$ of a new finite crack:

$$f\big(\sigma(x), \tau(x), \sigma_{\mathrm{c}}, \tau_{\mathrm{c}}\big) \geq 1 \quad \forall \quad x \in \Omega(\Delta a). \tag{7}$$

Cracks typically initiate from stress concentrations. Hence, stresses typically decrease with distance from the point of crack nucleation. At failure the material will only be overloaded in a small region around the point of crack nucleation. Hence, cracks admissible by the stress condition, Eq. (7), can only have a certain maximum length within the overloaded region. The stress

criterion constitutes an upper bound for the finite crack length $\Delta a$ as illustrated in the left half of Figure 2.

Requiring the simultaneous satisfaction of both the energy and the stress condition

$$\begin{cases} f\big(\sigma(x), \tau(x), \sigma_{\mathrm{c}}, \tau_{\mathrm{c}}\big) \geq 1 \quad \forall \quad x \in \Omega(\Delta a), \\ g\big(\overline{\mathcal{G}}_{\mathrm{I}}(\Delta a), \overline{\mathcal{G}}_{\mathrm{II}}(\Delta a), \mathcal{G}_{\mathrm{Ic}}, \mathcal{G}_{\mathrm{IIc}}\big) \geq 1, \end{cases} \tag{8}$$

yields a sufficient condition for crack onset. The concept is known as finite fracture mechanics and so-called coupled stress and energy criterion, Eq. (8), provides two equations to determine the two unknowns critical load and finite crack length. It is

physically sound and requires only the fundamental material parameters strength ($\sigma_{\mathrm{c}}, \tau_{\mathrm{c}}$) and toughness ($\mathcal{G}_{\mathrm{Ic}}, \mathcal{G}_{\mathrm{IIc}}$) as input. In particular, no assumptions on initial defects or numerical stabilization or regularization parameters are necessary (Weißgraeber et al., 2016).

Eq. (8) is a unified criterion for crack nucleation and crack growth. Crack initiation is governed by both conditions simultaneously and occurs as a finite crack increment $\Delta a > 0$. When a crack has formed, stresses at the crack tip are infinite.

For perfectly brittle materials the crack increment becomes infinitesimally small $\Delta a \to 0$ when such a crack tip singularity is present. Hence, continuous crack growth according to Griffith criterion of classical fracture mechanics is recovered.

This general observation also applies to skier-triggered weak layer collapse. Skier loading induces a stress concentration within the weak layer which may allow for the nucleation of finite sized anticracks provided the given loading satisfies both the stress and the energy condition. The subsequent stability of this initial anticrack is governed by the energy condition only.

## 2.3 Mixed-mode strength hypothesis

For the application of the general coupled stress and energy criterion, Eq. (8), to a particular material both the stress criterion $f$ and the energy criterion $g$ must be chosen accordingly to represent the material behavior. Skier-triggered weak layer collapse is governed by both shear and compression. A simple and common (for many engineering problems) interaction law determining an effective weak layer strength under mixed-mode loading is given by the quadratic interaction law

$$f_2(x) = \sqrt{\left(\frac{\sigma(x)}{\sigma_{\mathrm{c}}^-}\right)^2 + \left(\frac{\tau(x)}{\tau_{\mathrm{c}}}\right)^2}, \tag{9}$$

between the weak layer compressive strength $\sigma_{\mathrm{c}}^-$ and its shear strength $\tau_{\mathrm{c}}$. In Eq. (9) both strengths, $\sigma_{\mathrm{c}}^-$ and $\tau_{\mathrm{c}}$, are assumed constant and only compressive normal stresses $\sigma(x)$ are considered.

Failure of shear and compression loaded geological materials such as soil and rock is often modeled using the Mohr-Coulomb strength criterion

$$f_{\mathrm{mc}}(x) = \frac{\tau(x)}{\tau_{\mathrm{c}}^{\mathrm{mc}}(\sigma)}, \tag{10}$$

where the Mohr-Coulomb shear strength

$$\tau_{\mathrm{c}}^{\mathrm{mc}}(\sigma) = (\sigma_{\mathrm{c}}^+ - \sigma)\tan\phi, \tag{11}$$

depends on superimposed normal stresses $\sigma$, the tensile strength $\sigma_{\mathrm{c}}^+$ determined in pure uniaxial tension and the internal friction angle of the material $\phi$. Superimposed compression increases sustained shear loading.



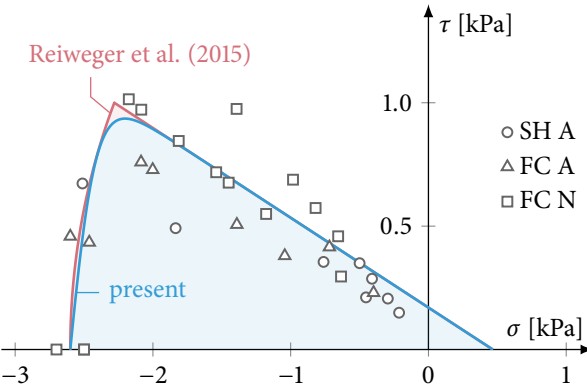

**Figure 3.** Smooth (present) and $C^0$-continuous (Reiweger et al., 2015) capped Mohr-Coulomb weak layer failure envelopes with experimental data on weak layers of natural surface hoar (SH N), natural faceted crystals (FC N) and artificial faceted crystals (FC A) taken by Reiweger et al. (2015). Properties chosen as given in Table 1.

Despite considering the influence of normal stress, the Mohr-Coulomb criterion, Eq. (10), is a pure shear criterion, i.e., it captures only the material's shear failure. Compressive failure is not considered as the corresponding shear strength, Eq. (11), increases indefinitely with superimposed compression. Modeling porous and collapsible media such as weak layers, additionally requires the introduction of a compressive strength. Based on their lab experiments Reiweger et al. (2015) propose to

introduce a compressive limit by capping the classical Mohr-Coulomb criterion using

$$f_{\mathrm{cap}}(x) = \frac{\tau(x)}{\tau_{\mathrm{c}}^{\mathrm{cap}}(\sigma)}, \tag{12}$$

where $\tau_{\mathrm{c}}^{\mathrm{cap}}$ is the shear strength in the cap region. It rapidly decreases from its maximum as to zero with superimposed compression. In the notation of the present work the equation proposed by Reiweger et al. (2015) reads

$$\tau_{\mathrm{c}}^{\mathrm{cap}}(\sigma) = \beta \sqrt{1 - \frac{\left(\sigma + \sigma_{\mathrm{c}}^{+}\right)^2}{\left(\sigma_{\mathrm{c}}^{-} + \sigma_{\mathrm{c}}^{+}\right)^2}}, \tag{13}$$

with

$$\beta = \tau_{\mathrm{c}}^{\mathrm{max}} \sqrt{\frac{\left(\sigma_{\mathrm{c}}^{+} + \sigma_{\mathrm{c}}^{-}\right)^2}{\left(\sigma_{\mathrm{c}}^{+} + \sigma_{\mathrm{c}}^{-}\right)^2 - \left(\frac{\tau_{\mathrm{c}}^{\mathrm{max}}}{\tan\phi}\right)^2}}, \tag{14}$$

where $\tau_{\mathrm{c}}^{\mathrm{max}}$ is the absolute maximum shear strength. It corresponds to the intersection of the cap, Eq. (12), and the Mohr-Coulomb criterion, Eq. (10). Again, $\sigma_{\mathrm{c}}^{+}$ and $\sigma_{\mathrm{c}}^{-}$ are the tensile and compressive strengths measured in pure uniaxial tension and compression, respectively. Hence, the effective capped Mohr-Coulomb weak layer shear strength is given as the minimum

of the Mohr-Coulomb shear strength $\tau_{\mathrm{c}}^{\mathrm{mc}}$ and the cap shear strength $\tau_{\mathrm{c}}^{\mathrm{cap}}$

$$\tau_{\mathrm{c}}^{\mathrm{cmc}}(\sigma) = \min\left\{\tau_{\mathrm{c}}^{\mathrm{mc}}(\sigma), \, \tau_{\mathrm{c}}^{\mathrm{cap}}(\sigma)\right\}. \tag{15}$$



Compressive failure occurs when normal stresses reach the compressive strength $\sigma = \sigma_{\mathrm{c}}^{-}$. At this point the cap shear strength reduces to zero $\tau_{\mathrm{c}}^{\mathrm{cap}}(\sigma_{\mathrm{c}}^{-}) = \tau_{\mathrm{c}}^{\mathrm{cmc}}(\sigma_{\mathrm{c}}^{-}) = 0$ and the capped Mohr-Coulomb criterion

$$f_{\mathrm{cmc}}(x) = \frac{\tau(x)}{\tau_{\mathrm{c}}^{\mathrm{cmc}}(\sigma)}, \tag{16}$$

is always satisfied ($f > 1$). Likewise tensile failure occurs when $\sigma = \sigma_{\mathrm{c}}^{+}$ because the Mohr-Coulomb shear strength vanishes
$\tau_{\mathrm{c}}^{\mathrm{mc}}(\sigma_{\mathrm{c}}^{+}) = \tau_{\mathrm{c}}^{\mathrm{cmc}}(\sigma_{\mathrm{c}}^{+}) = 0$. The capped Mohr-Coulomb criterion proposed by Reiweger et al. (2015), Eq. (16), is shown in Figure 3 together with their experimental data on three different weak layers. The criterion provides a proper failure envelope for all investigated weak layers comprising natural surface hoar as well as natural and artificial faceted crystals. However, it kinks at the transition between classical Mohr-Coulomb criterion and cap which can cause problems in optimization procedures.

In order to simplify the mathematical formulation of the cap and to provide a smooth envelope we propose a new effective
weak layer shear strength

$$\tau_{\mathrm{c}}^{\mathrm{scmc}}(\sigma) = \tanh\left(\omega\left(\sigma - \sigma_{\mathrm{c}}^{-}\right)\right)\tau_{\mathrm{c}}^{\mathrm{mc}}(\sigma), \tag{17}$$

where $\omega$ controls the sharpness of the transition into the cap region and hence the maximum shear strength. Here we use $\omega = 5$. The corresponding stress criterion reads:

$$f_{\mathrm{scmc}}(x) = \frac{\tau(x)}{\tau_{\mathrm{c}}^{\mathrm{scmc}}(\sigma)}. \tag{18}$$

This criterion is visualized in Figure 3 alongside the formulation proposed by Reiweger et al. (2015) and the corresponding experimental data. Both failure envelopes represent the given test data well.

## 2.4 Mixed-mode energy criteria

Mixed-mode energy criteria describe the interaction of crack opening modes I, II and III. Mode I corresponds to crack opening normal to the crack faces which comprises both tearing and collapse each associate to a distinct fracture toughness, $\mathcal{G}_{\mathrm{Ic}}^{+}$ and
$\mathcal{G}_{\mathrm{Ic}}^{-}$, respectively (see part I). Modes II and III are shear crack modes and correspond to displacements tangential to the crack faces. The former originates from in-plane shear loading, the latter from out-of-plane shear loading. A crack traveling upslope or downslope is driven by mode I and mode II energy release rates. Cross slope cracks are mode I and mode III driven. Because PSTs are typically cut upslope, crack propagation is governed by a certain interaction of mode I and II depending on slope angle and slab thickness and no mode III contributions are present. Both shearing modes II and III are often considered to
originate from similar physical mechanisms and associated to similar toughness values $\mathcal{G}_{\mathrm{IIc}} \approx \mathcal{G}_{\mathrm{IIIc}}$.

For the interaction of compressive mode I and mode II loading the following simple general mixed-mode energy criterion

$$g(\Delta a) = \left(\frac{\mathcal{G}_{\mathrm{I}}(\Delta a)}{\mathcal{G}_{\mathrm{Ic}}^{-}}\right)^{n} + \left(\frac{\mathcal{G}_{\mathrm{II}}(\Delta a)}{\mathcal{G}_{\mathrm{IIc}}}\right)^{n}, \quad n \in [1, \infty) \tag{19}$$

may be used. The exponent $n$ characterizes the strength of the interaction. It is linear for $n = 1$ and vanishes as $n \to \infty$ where failure is governed purely by one mode or the other. As a simple and consistent choice we use $n = 1$ throughout the present
work. In general, the shape of the strength criterion and the mixed-mode energy criterion are independent and do not depend.





Other mixed-mode criteria typically used in engineering applications such as the classical criteria by Hutchinson and Suo (1991) or Benzeggagh and Kenane (1996) are not suitable for weak layer collapse. Both are only applicable when the mode I and mode II toughnesses are of the same order. As discussed in part I, the tearing mode I ($\mathcal{G}_{\mathrm{Ic}}^{+}$) and shear mode II ($\mathcal{G}_{\mathrm{IIc}}$) fracture toughnesses are of the same order of magnitude. However, as discussed in part I, the collapse mode I fracture toughness $\mathcal{G}_{\mathrm{Ic}}^{-}$

is up to two orders of magnitude larger than $\mathcal{G}_{\mathrm{Ic}}^{+}$. Hence, the above classical criteria are not applicable to weak layer collapse because of its different micromechanical failure mechanism.

### 2.5   Solution of the coupled criterion

As discussed in section 2, neither an exclusive stress criterion nor an exclusive fracture mechanics criterion are sufficient to describe anticrack nucleation. Both are necessary conditions for failure and must be coupled in some way to obtain a sufficient

condition for failure. In this work we use the above discussed mixed-mode criteria for compressive and shear stresses as well as mode I and II energy release rates, to establish a coupled mixed-mode criterion in the framework of finite fracture mechanics. Using the closed-form analytical solutions for deformations, stresses and energy release rates of a slab on a deformable weak layer presented in part I of this work, allows for very efficient evaluations of the individual mixed-mode criteria.

Figure 4 shows the principle implementation of the coupled stress and energy failure criterion. Initially, the load at failure as

well as the size of initiated crack are unknown. The general normal and shear stress solution of the mechanical model are used to calculate the mixed-mode stress criterion as a function of distance from the skier. Analogously, the solutions for mode I and II energy release rates are used to compute the mixed-mode energy condition as a function of the initially unknown finite crack length. Now the most critical situation satisfying the two conditions simultaneously must be determined. It is associated to the lowest critical extra load. This yields an optimization problem with two free (initially unknown) variables: the critical load and

the size of the initiated crack:

$$
\begin{aligned}
F_{\mathrm{f}} = \min_{F, \Delta a} \Big\{ &F \mid f\big(\sigma(x), \tau(x), \sigma_{\mathrm{c}}, \tau_{\mathrm{c}}\big) \geq 1 \ \ \forall \ x \in \Omega(\Delta a) \\
&\wedge \ g\big(\overline{\mathcal{G}}_{\mathrm{I}}(\Delta a), \overline{\mathcal{G}}_{\mathrm{II}}(\Delta a), \mathcal{G}_{\mathrm{Ic}}^{-}, \mathcal{G}_{\mathrm{IIc}}\big) \geq 1 \Big\}.
\end{aligned}
\tag{20}
$$

The smallest load $F$ satisfying both individual criteria $f$ and $g$ is to be found for any kinematically admissibile finite crack $\Delta a$. This optimization problem can be treated with standard minimization schemes.

The present implementation of the coupled criterion provided as supplementary material requires up to $5\,\mathrm{s}$ of computation time on a standard desktop computer. However, the coupled criterion is suited for parallel computing and the computational cost can be reduced significantly using an iterative solution scheme (Felger et al., 2019) adapted to the present mixed-mode criteria. This way, the computation time can be reduced to a few milliseconds.

### 3   Results

In the following we use the mechanical model derived in part I and the weak layer failure criterion proposed in section 2 to discuss effects of important snowpack properties on skier-triggered slab avalanche formation. In each study slabs loaded by




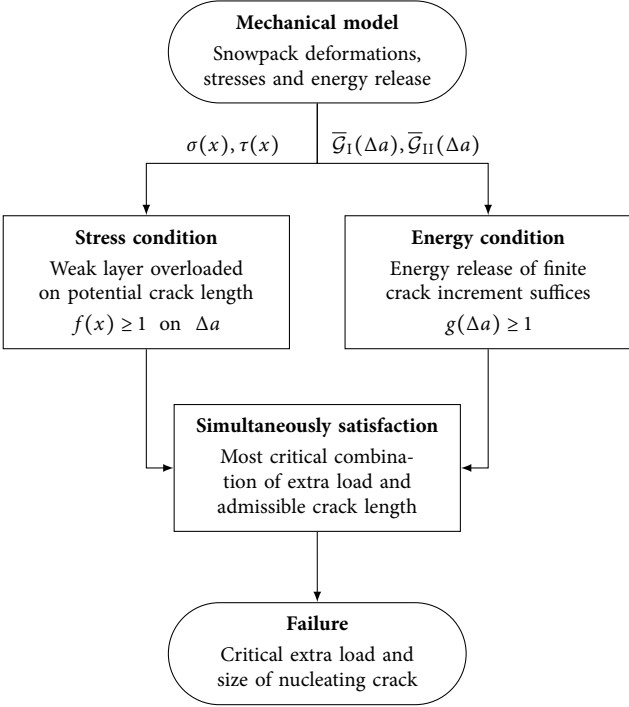

**Figure 4.** Solution of the coupled stress and energy criterion for anticrack nucleation. The model presented in the first part of this work provides required stresses and energy release rates. Using the given closed-form analytical solutions, the minimum critical skier loading satisfying both conditions can be identified efficiently employing optimization procedures.

their own weight plus an additional local skier load are considered. The given failure criterion predicts the critical extra load which initiates failure within the weak layer.

It is important to point out that the finite crack length $\Delta a$ and the critical skier load $F$ shown in the following are direct results of the present model and explicitly linked. They cannot be considered or studied individually. The pair of both quantities

5   is a unique solution to the coupled stress and energy criterion, Eq. (8). The critical load is the lowest possible load required to nucleate an initial anticrack of size $\Delta a$. The size of the initial crack is different from the critical crack length $a_\mathrm{c}$ as used, e.g., by Gaume and Reuter (2017). $\Delta a$ does not represent the critical crack length for crack propagation. It is the size of initial weak layer collapse owing to overcritical skier loading. Whether this crack $\Delta a$ will propagate is determined by the Griffith criterion $\mathcal{G} = \mathcal{G}_\mathrm{c}$.

10   We consider the slope angle $\varphi$, the thickness of the slab $h$ resting on the weak layer, the slab's density $\rho$ and the slab's Young's modulus $E$. For static skier loading the local load is $F = mgb/l_0$. Additionally, different mixed-mode failure criteria and the effect of the slope of the fracture envelope are studied. Each study examines the impact of one individual parameter. All other properties are chosen as given in Table 1. If not stated otherwise, the Young's modulus of the slab is calculated from





**Table 1.** Material properties used in parametric studies of the finite fracture mechanics criterion.

| Property | Symbol | Value |
|---|---|---|
| Slope angle | $\varphi$ | $30\,^\circ$ |
| Slab thickness[*] | $h$ | $40\,\mathrm{cm}$ |
| Weak layer thickness[*] | $t$ | $2\,\mathrm{cm}$ |
| Effective ouf-of-plane ski length | $l_\mathrm{o}$ | $1\,\mathrm{m}$ |
| Young's modulus slab | $E_\mathrm{slab}$ | $4\,\mathrm{MPa}$ |
| Young's modulus weak layer | $E_\mathrm{weak}$ | $0.15\,\mathrm{MPa}$ |
| Poisson's ratio slab & weak layer | $\nu$ | $0.25$ |
| Slab density | $\rho$ | $200\,\mathrm{kg m^{-3}}$ |
| Tensile strength[†] | $\sigma_\mathrm{c}^+$ | $0.4\,\mathrm{kPa}$ |
| Compressive strength[†] | $\sigma_\mathrm{c}^-$ | $2.6\,\mathrm{kPa}$ |
| Shear strength[†] | $\tau_\mathrm{c}$ | $0.7\,\mathrm{kPa}$ |
| Mode I fracture toughness[†] (collapse) | $\mathcal{G}_\mathrm{Ic}^-$ | $3\,\mathrm{Jm^{-2}}$ |
| Mode II fracture toughness[†] (shear) | $\mathcal{G}_\mathrm{IIc}$ | $0.15\,\mathrm{Jm^{-2}}$ |

[*] Thicknesses measured slope normal.

[†] Strength and fracture toughness parameters are properties of the weak layer.

density $\rho$ using an empirical power law fit to the data of Scapozza (2004) in plane strain conditions

$$E = \frac{1}{1-\nu^2}\,5.07 \times 10^3 \left(\frac{\rho}{\rho_0}\right)^{5.13}\mathrm{MPa}, \tag{21}$$

with the density of ice $\rho_0 = 917\,\mathrm{kg\,m^{-3}}$. Except for the specific analysis of different stress criteria, we employ $f_2$, Eq. (9), as the stress criterion because of its simplicity.

## 3.1  Effects of physical properties

Figure 5 shows the effect of the slope angle $\varphi$ on the critical extra load $F$. The critical loading decreases significantly with increasing slope inclination. Because of the interaction of compression and shear deformation, and because of the interaction of stress and energy, the change is nonlinear. The length of the finite anticrack $\Delta a$ initiated at the critical loading increases only moderately with increasing slope angle. The size of the initiated cracks lies between 30 and 80 cm. The thin slab of $h = 20\,\mathrm{cm}$ leads to lower critical loads in flat terrain. However, for steeper slopes this effect is reversed and the critical load of the thicker slab $h = 40\,\mathrm{cm}$ falls below that of the thin slab.

The slab thickness directly affects the critical load. Figure 6 shows a study of this parameter for slab thicknesses up to 90 cm. For low to moderate slab thicknesses the critical extra load sustained by the snowpack increases. On the considered slope angles of $20°$ and $30°$ the critical load decreases above 30 and $40\,\mathrm{cm}$ slab thickness, respectively. The size of the initial cracks increases monotonously. As observed in Figure 5, steeper slopes sustain smaller critical loads.





**Figure 5.** Effect of the slope angle $\varphi$ on the critical additional loading $F$ on a defect-free snowpack (solid line) and on the length of predicted initial finite anticrack $\Delta a$ (dotted line). Steeper slopes cause a significant reduction of the critical skier loading and a moderate increase of the crack lengths.





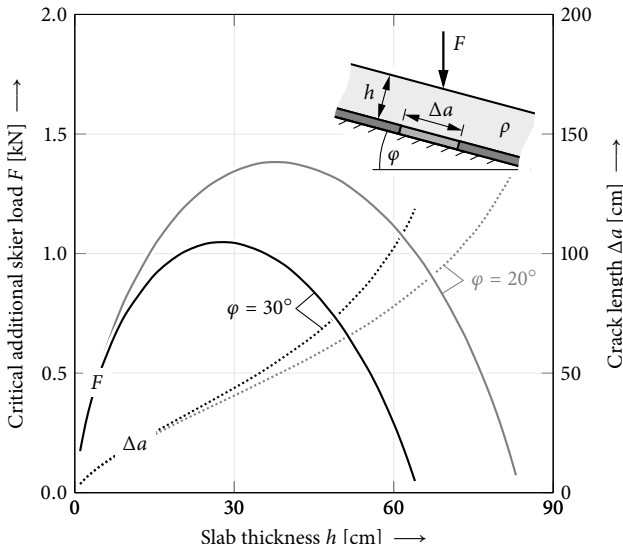

**Figure 6.** Critical loading of a snowpack without assumed defects and the size of initiated anticrack as a function of slab thickness. Thicker slabs transfer concentrated loads more uniformly allowing for larger point loads. Above a certain thickness, failure is dominated by the slab's own weight reducing admissible additional loads until self-release occurs. The finite anticrack length increases continuously with increasing slab thickness.

Figure 7 shows the impact of both slab density $\rho$ (black lines) and Young's modulus $E$ (gray line). Increasing only the Young's modulus stiffens the slab, distributes the skier load more evenly across the weak layer and intuitively increases the critical extra loading. The influence of the slab density $\rho$ is examined both for a constant Young's modulus, $E = \text{const.}$, independent of slab density and for the more realistic case of a density-dependent Young's modulus, $E = E(\rho)$, according to

Eq. (21). Increasing only the density (at a constant Young's modulus) increases the slab weight and hence reduces the critical skier loading. If the slab's Young's modulus $E$ depends on slab density, denser slabs are stiffer and distribute loads more evenly. As observed for the slab thickness, the initial stiffening allows for larger skier loads. However, at a critical density slab weight increase dominates failure and critical skier loads drop quickly. In either case, finite anticrack lengths increase with slab density (not shown in Figure 7). As the critical skier load decreases, the energy release rate at critical loading reduces. This requires

longer cracks in order to release sufficient energy for crack formation. For density-dependent Young's moduli the stiffening makes this effect even more pronounced.

## 3.2   Comparison of mixed-mode criteria

The coupled stress and energy criterion accounts for shear and compressive failure. In the following the effect of the mixed-mode criteria for stress and energy will be discussed.

The choice of the mixed-mode fracture envelope and the ratio of the mode I and mode II fracture toughness is studied in Figure 8. The results of the solution of the implicitly coupled criterion is shown for different fracture toughness ratios.





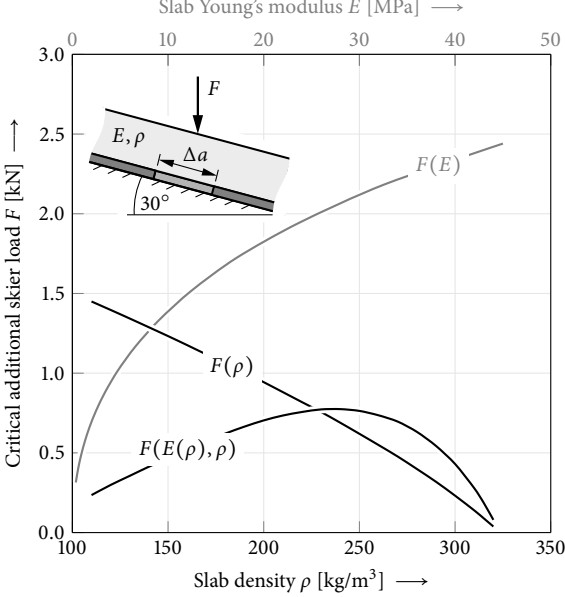

**Figure 7.** Effect of slab density $\rho$ on critical additional skier loading $F$ of a defect-free snowpack and the corresponding length of the initiated crack $\Delta a$. If the slab's Young's modulus is independent of slab density the increasing slab weight reduces the critical skier loading. A density-dependent Young's modulus according to Eq. (21) stiffens denser slabs which initially dominates failure and increases the critical skier load. Above a certain threshold, the weight gain becomes dominant and the critical skier loads reduces distinctly. The size of the initiated cracks $\Delta a$ increases with slab density.

For all fracture toughness ratios the critical skier load decreases monotonously with slope angle. With the chosen parameters natural release is predicted on slopes steeper than $\varphi \approx 60°$. For a mode I/mode II fracture toughness ratio $\mathcal{G}_{\mathrm{IIc}}/\mathcal{G}_{\mathrm{Ic}}^- > 1/30$ the corresponding length of the initiated finite cracks increases. When $\mathcal{G}_{\mathrm{IIc}}$ is below this threshold the finite crack length decreases.

The effect of the criterion for the interaction of shear and normal stress in the weak layer is studied in Figure 9. The simple
5   quadratic stress interaction $f_2$, Eq. (9), is compared against the capped Mohr-Coulomb criterion in its smooth formulation $f_{\mathrm{scmc}}$, Eq. (18). The individual constituents of the capped Mohr-Coulomb – the Mohr-Coloumb $f_{\mathrm{mc}}$, Eq. (10), and the capping function $f_{\mathrm{cap}}$, Eq. (12) – are shown as well. The capping criterion $f_{\mathrm{cap}}$ corresponds to weak layer failure due to normal stress only and shows slightly decreasing critical skier loading with increasing slope angle. The Mohr-Coloumb criterion for shear loading of the weak layer $f_{\mathrm{mc}}$ shows a strong dependence on the slope angle. The corresponding critical load becomes infinite
10   in flat terrain and vanishes on slopes steeper than $45°$ for the present set of parameters. The capped Mohr-Coulomb criterion $f_{\mathrm{scmc}}$ combines both criteria with a sharp transition at approximately $20°$. The simple quadratic stress interaction criterion $f_2$ describes a similar trend but provides a smooth transition of the failure load with slope angle. At small inclinations the effect of the slope angle is more pronounced than for the capped Mohr-Coulomb. Yet, for the chosen parameters the critical skier load determined using $f_2$ does not vanish on steep slopes.



**Figure 8.** Comparison of mixed-mode energy criteria for a defect-free snowpack loaded by a point-load and the weight of the slab above the weak layer.





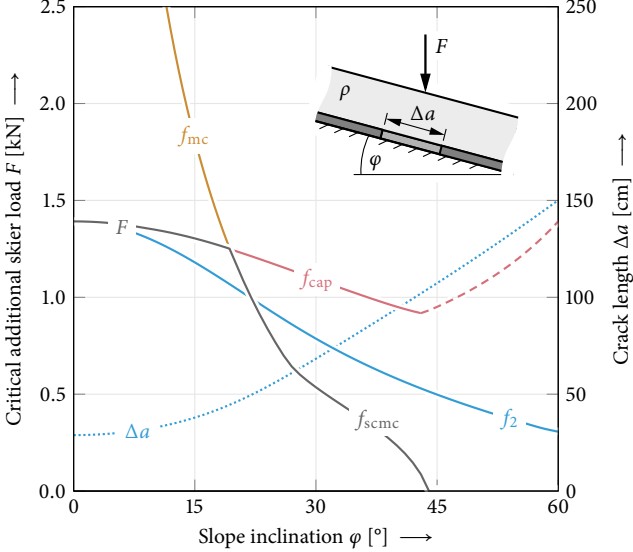

**Figure 9.** Comparison of mixed-mode stress criteria for a defect-free snowpack with slab thickness $h = 20\,\text{cm}$. The quadratic interaction criterion $f_2$, the smooth capped Mohr-Coulomb criterion $f_{\text{scmc}}$ as well as its constituents $f_{\text{mc}}$ and $f_{\text{cap}}$ are shown.

## 4 Discussion of anticrack nucleation criterion

In the following results of the parametric studies given above are discussed in order to elucidate basic features of the present failure criterion.

The coupled stress and energy criterion employed in the present work uses linear-elastic solutions provided in part I of this
work and the failure condition itself does not account for time-dependence. Hence, any loading which occurs much slower than that of a descending or ascending skier may not be captured correctly. This applies in particular to natural avalanche release due to increased weight of the slab or thermal effects (Jamieson and Johnston, 1999). In these cases, the strain rates are much lower ($< 10^{-7}\,\text{s}^{-1}$) (McClung, 1979) and the effect of temporal effects like sintering must be accounted for in an extended failure criterion (van Herwijnen and Miller, 2013; Birkeland et al., 2019a). Since, material properties of snow strongly depend
on strain rate (Reuter et al., 2019), they must be chosen to correctly represent the rapid loading by a skier and the corresponding brittle failure behavior.

Local overloading (stress exceeding strength) and sufficient energy release are only necessary conditions of failure (Leguillon, 2002). This was also addressed by Gaume and Reuter (2017). They use Föhn's solution to calculate skier-induced shear stress in the weak layer and to determine the length of the overloaded weak layer. This length is then compared to a critical
crack length required for crack propagation proposed by Gaume et al. (2017) to assess the stability of the snowpack. To study layered slabs they use linear elastic finite element analyses to determine the size of the overloaded weak layer. In the present work, we employ the framework of finite fracture mechanics. The concepts directly links the two necessary conditions to provide a coupled stress and energy criterion as the sufficient condition for brittle crack onset. Because the length of the initiated





crack is a direct result of the solution of the two implicitly coupled equations it does not have to be assumed. To be clear, we do not require the assumption of, e.g., super-weak zones or initial flaws within the weak layer. Studying mixed-mode stress and energy criteria (Figures 8 and 9) we show that mixed-mode weak layer loading and mixed-mode energy release rate has an important impact on the critical loading of a snowpack.

The present model extends the concept of anticracks by combining strength and energy as coupled conditions for the nucleation of anticracks. Physical interaction criteria of shear and normal stress as well as the mixed-mode energy release are covered. The anticrack model by Heierli extended the understanding of avalanche release and included the remote-triggering of avalanches and *whumpf*-sounds for which shear failure models cannot provide physical explanations. However, since the model only considers the weak layer's fracture toughness but not its compliance important parameter dependences like the

effect of slope angles contradict observations (Gaume et al., 2017). The present model considers the energy release due of collapse as well as shear in the condition for initial failure. Collapse is not considered as a secondary process as postulated by Reiweger et al. (2015) because normal deformation is directly linked to the initiation of weak layer failure.

The effect of slope angle $\varphi$ (Figure 5) shows a monotonous decrease of the critical load on the snowpack with increasing slope angles. The critical load for $\varphi = 0°$ is finite and corresponds to weak layer failure in flat terrain causing *whumpfs* and

under certain circumstances remote triggering of avalanches. The critical skier load vanishes in very steep terrain. Here, natural release is expected for the chosen material properties. The size of initiated cracks increases as the dominant mode I energy release rate reduces and longer cracks are required to satisfy the energy condition. Comparing the two studied slab thicknesses, several mechanisms are interacting. Stronger bridging of thicker slabs increases the length of initiating cracks. This causes higher sustained skier loads in flat terrain. The lower strength of the weak layer in shear governs the decrease of the critical

loading with slope angle. This finally leads to natural release which occurs of course earlier for thicker and thus heavier slabs.

Figure 6 shows the impact of the slab thickness $h$. Thicker slabs transfer the skier load more uniformly and increase the critical loading $F$. Above a certain slab thickness, the increasing slab weight dominates failure reducing the critical skier loading until self-release occurs. Modeling the skier loading as a concentrated force yields vanishing critical loads if no slab is present. A more realistic result would be obtained modeling skier loading as a distributed load. However, this would only

be relevant for very thin slabs. As thicker slabs distribute loads more evenly an increase in slab thickness is accompanied by a significant increase of the finite anticrack length $\Delta a$. It is the bridging effect discussed in part I which leads to much longer initial cracks for increased slab thicknesses. Since shear loading increases on steeper slopes the overall magnitude of the critical loads decreases with increasing slope angle. However, the characteristic slab thickness dependence remains unchanged. Slab thickness directly affects the possibility of initiating cracks in flat or inclined terrain. In particular, thin slabs allow for easy

triggering of weak layer failure. In order to comprehensively account for the effect of the slab thickness, material and failure parameters should be considered as functions of the slab thickness (Bažant et al., 2003).

The effect of slab density $\rho$ and Young's modulus of the slab $E$ are studied in Figure 7. Increasing the Young's modulus of the slab increases bridging and hence, the capability of distributing the load on a wider area the weak layer. Considering time-dependent slab stiffening explains why the stability of snowpacks can increase over time. With increasing weight load

of the slab owing to increased slab density the critical load decreases monotonously. If now the Young's modulus is assumed





density-dependent by means of the empirical relation of Scapozza, Eq. (21), these two effects compete. For small slab densities the effect of increasing Young's modulus and accordingly bridging dominates leading to increasing critical loads. However, for higher slab densities the effect of the extra weight outweighs bridging and the failure load decreases. Eventually natural release is predicted.

The mixed-mode law for the interaction of energy release rates of compressive (mode I) and shear (mode II) deformation of the initiated crack are studied in Figure 8. Mode-mixity is studied by varying the magnitude of the mode II fracture toughness $\mathcal{G}_{\mathrm{IIc}}$ relative to a constant mode I fracture toughness $\mathcal{G}_{\mathrm{Ic}}$. For flat terrain the contribution of shear deformation in the weak layer is negligible and hence, the parameter variation of $\mathcal{G}_{\mathrm{IIc}}$ has no effect on the failure load at $\varphi = 0$. It is discussed in part I of the present work, that the fracture toughness in mode I is significantly higher than that of shear, as the corresponding microstruc-

tural failure in compression dissipates much more energy than simple shearing failure of a porous weak layer structure. Hence, if the mode II fracture toughness is chosen within the order of magnitude of the mode I fracture toughness, a pronounced increase of the length initiated finite cracks is predicted as the slopes inclination increases. On steeper slopes failure is dominate by mode II. If the corresponding fracture toughness $\mathcal{G}_{\mathrm{IIc}}$ is chosen too large, unrealistically long cracks are required to satisfy the energy condition. Crack lengths agree qualitatively with predictions of other models when $\mathcal{G}_{\mathrm{IIc}}$ is 20 to 40 times smaller

than $\mathcal{G}_{\mathrm{Ic}}$ (Gaume et al., 2017).

Figure 9 shows effect of the given interaction criteria of shear and normal stress in the weak layer. In flat terrain lateral shear stresses vanish and using the Mohr-Coulomb criterion only would predict infinite failure loads. Therefore, the capped Mohr-Coulomb has been introduced (Reiweger et al., 2015). However, the cap criterion itself shows a peculiar effect when used within coupled stress and energy criterion. Despite decreasing normal stresses with increasing slope angle, the coupled criterion

yields slightly decreased critical skier loading. This occurs because of increasing mode II energy release rate contribution to the fulfillment of the energy condition. This implicitly allows for shorter initiated cracks and lower failure loads. With a chosen constant shear strength, $\tau_{\mathrm{c}} = 0.7\,\mathrm{kPa}$, the quadratic interaction criterion $f_2$ does not render the effect of very low shear strengths when low normal loading occurs. Therefore, the decrease of the critical load with increased slope angles is much less pronounced than with the Mohr-Coulomb criterion. This has been addressed by Reiweger et al. (2015). They pointed out that

for steeper slopes the Mohr-Coulomb criterion is the relevant stress criterion and the cap can be neglected. In the present model the transition from cap criterion to Mohr-Coulomb occurs near $23°$. It is to point out that our model considers lateral weak layer shear stress, i.e., shear stress owing to lateral displacements of the slab only. Transverse shear stress, i.e., shear stress originating from shear deflection of the slab is not accounted for. The latter is caused by skier loading even in flat terrain and could contribute to lower failure loads predicted by the pure Mohr-Coulomb criterion. However, the former is more relevant

for weak layer failure and dominates in particular on inclined slopes.

The present work studies the onset of weak layer failure. For a subsequent release of an avalanche the growth of this crack and the trough-thickness fracture of the slab release are decisive as well. If no slab fracture occurs only a settlement of the snowpack occurs (*whumpf* sound). If the strength of the slab is very low shooting cracks around the point of local loading occur relaxing slab stresses and no propagation of the anticrack is possible. Crack growth is directly controlled by the stability

of the initiated and subsequently growing crack. The case of infinitesimal crack growth is covered by Griffith's criterion of





linear elastic fracture mechanics (LEFM) and depends only on differential energy release rates in compression and shear. The present model also provides these quantities as an outcome of the model in part I and in conjunction with the mixed-mode energy criterion (Eq. 19) the stability of the initiated cracks can be assessed. Once cracks grow long slab touchdown can occur. This limits the energy release rates of long cracks. Hence, for certain snowpacks no propagation of cracks to a critical size

is possible as either the energy release of long cracks is insufficient and the crack stops or the slab itself is to weak and fails close to the initiated weak layer failure. The stability of cracks under thin slabs and their propagation in flat terrain has been addressed by Gauthier and Jamieson (2008b).

Slab failure is typically induced by a combination of local bending and tension loading of the slab leading to slab fracture (McClung and Borstad, 2019). Studies of fracture initiation indicate that the slab fractures in a brittle manner (Bair et al.,

2016). In future the present model may be extended to account for fracture of the (layered) slab as a potential consequence of anticrack nucleation and growth.

Since, the propagation of existing cracks is purely energy-controlled (Anderson, 2017), also the crack speed is only controlled by the energy balance, i.e., in this case the energy balance of energy release at the crack front and the energy required for crack growth. As the latter is just the fracture toughness, measurements of the crack speed could be a way to determine the

value of the fracture toughness. However, in inclined terrain cracks will always expand under mixed-mode conditions. Hence, the evaluation of the crack speed measurements must happen with a model that is able to provide the energy release rates of mode I and mode II.

Effects of the slab and the fracture behavior of the weak layer will always interact, causing a complex failure behavior. Good models may allow for separating effects and thus allow for better understanding of field experiments. For instance,

the critical cut length in PST experiments may correlate with slab thickness but does not directly depend on it. Further, Bair et al. (2014); van Herwijnen et al. (2016) show that edge effects in PSTs can have an important effect. Hence, either field experiments must be modified or models need to be established which can account for such edge and size effects. Similar considerations are necessary for Rutschblock tests. The optimal setup is for this skier load stability test must be assessed and it is to determine which properties can be deduced from this test. The recent work by Birkeland et al. (2019b) proposes a new

way to conduct PSTs. It suggests to change the geometry of one weak layer and slab configuration by adding slab thickness. Such an experiment could be of high relevance for identifying the mixed-mode fracture toughness of the particular weak layer.

The present model bases on closed-form equations of the slab and the weak layer displacement fields. The proposed failure criterion for nucleation of anticrack makes use of this model and solves the implicit equations of the coupled criterion with high efficiency. Since the computational effort is much smaller than for numerical models, the present model can be used readily in

large parameter studies or uncertainty quantification analyses.

## 5 Conclusions

A novel criterion for anticrack nucleation has been proposed on the basis of the closed-form analytical solution proposed in part I of this work.



1. The criterion implicitly links a stress criterion for local overloading of the weak layer with a global fracture mechanics criterion of the energy balance of crack initiation.

2. It is shown that in order to study weak layer failure the interaction of shear and compression stresses and mixed-mode energy release rates must be considered. Failure is governed by both strength and fracture toughness properties of the weak layer.

3. Parametric studies show that the proposed failure criterion is able to correctly render physical effects observed in slab avalanche release or field tests.

4. The model can be the basis for further analysis of mixed-fracture of the weak layer, propagation of weak layer failure and the failure of the slab above the weak layer.

*Code availability.* The analysis code of both the modeling framework in part I and the mixed-mode failure criterion based on this framework will be published in an online GitHub repository for public access.

*Author contributions.* Both authors defined the scope of the work and developed the present failure model together. PW provided most of the introduction and the discussion. PLR conducted the parametric studies. PLR and PW wrote the final manuscript with equal contribution.

*Competing interests.* The authors declare that they have no conflict of interest.

*Acknowledgements.* We would like to thank Alec van Herwijnen and Johan Gaume for detailed discussion of the present work and the current understanding of the physics of slab avalanche releas. We want to thank Karl Birkeland, Bastian Bergfeld and Ned Bair for the interesting exchange on snow stability and modeling of weak layer failure. We acknowledge support by the German Research Foundation and the Open Access Publishing Fund of Technische Universität Darmstadt.





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
