# Peer review of "Modeling snow slab avalanches caused by weak layer failure – Part II: Coupled mixed-mode criterion for skier-triggered anticracks"

_The Cryosphere, 2019_

## Referee Comment (RC1) · Michael Zaiser (Referee) · 2 Jul 2019

The paper deals with the formulation of a failure criterion for collapse-like failure of snow and application to skier triggering of avalanches. It combines a fracture mechanical and a strength-of-materials approach to forumlate a criterion for weak layer failure that does not rely on assuming a pre-existing flaw. Fracture mechanical and strength-of-materials ('snow stability index') approaches to snow failure have both been used in the snow literature, and sometimes in a manner that confounds their respective domains of application. Thus a unification of sorts is an inherently desirable undertaking.

[Figure]

The authors first discuss contradictions that may arise from an uncritical application of either strength-of-materials or fracture mechanical criteria to situations where those criteria do not apply. A nice example is given by Eqs. (3) and (4) which are particularly instructive as I have encountered very similarly flawed reasoning in a recent manuscript under review for The Cryosphere: Basically, one cannot simply apply an energy argument to an uncracked specimen and deduce a failure stress from it, nor can one invert the argument and convert a failure stress into a fracture toughness by evaluating the overall energy stored within the sample. By contrast, I find the argument surrounding Figure 1 less convincing. Reference is made to Bazant 1984 but, if we adopt Bazant's reasoning I see no reason why the strength of an uncracked specimen as shown in Figure 1 should be size dependent (in fact, Bazant's relation predicts strength to be size independent at small sample sizes whatever the crack length).

The FFM criterion attempts to resolve the well-known conundrum that exists in theories of fracture, namely that the transition from stress-induced damage accumulation (no crack) to the propagation of a critical crack is not well understood. It does so by combining a strength of materials criterion to obtain an upper estimate of the crack length (a crack can at maximum form over a length where the stress is above sigma_c) with a lower estimate (the ensuing crack must be able to propagate). The crack forms if the lower bound falls below the upper one.

Taking the example provided by the authors makes me wonder whether I understand the criterion correctly. Consider the tensile beam of Eqs. (3),(4) of the manuscript. Load this beam at the stress sigma_c such that the stress based criterion for mode I failure is fulfilled over the entire cross section. Finite fracture mechanics seems to indicate that all energy stored in the beam is released if a cross-section spanning crack is formed. If that is true, however, then Eq. (3) entails a critical beam length l $\sim$ (2EG_c/sigma_c^2) below which the beam cannot fail because that energy is insufficient. Now I insert typical values of steel, say KIc = 40 MPa m^1/2, sigma_c = 500 MPa, to obtain with EG_c $\sim$ KIc^2 a critical length in the range of 1 cm. Ehem. Do the

authors want to imply that my 1cm tensile sample is too small to break??????? Clearly I must be misunderstanding something.

I have a second objection. Consider again a thin tensile beam with thickness a and length l » a loaded at a stress slightly above sigma_c. Let us now assume the energy criterion is fulfilled: sigma_cˆ2 l > 2 E G_c. So the beam breaks. Now consider the same beam but embedded as surface fiber into a bending beam as in Figure 1. Let the bending moment be such that a region of thickness a from the surface is above the critical stress. In that case, the energy release will be less than for the free standing beam, and it may well be that the crack cannot form since sigma_cˆ2 a < 2 E G_c. However, the stress state in the considered volume is identical in both cases. The only thing which the volume elements (and the microstructure, grains, dislocations, atoms......) in the beam know about the outside world is the local stress acting on them. How do they understand that, in the first case, they should form a crack instantaneously, and in the second case, not?

I kindly request the authors to clarify the above two points.

Once we accept the basic approach, the development of stress and energy based failure criteria looks sensible. Concerning the skier loading, I have a question since the loading model is not very clearly described. I presume the authors assume plane strain conditions in which case F would need to be a load per unit length. On line 11, page 11 there is a mysterious b which seems not defined. As to material parameter choices, I commented on that point in relation to the companion paper.

A minor point: It is noted as an inherent flaw of models that assume subcritical damage accumulation that 'such models would predict avalanche release if only enough skiers ski the same slope in close temporal succession'. While such may not be generic behavior, I cannot see why this should be impossible to happen. I know of several passages in the avalanche literature reporting that several skiers may ski a slope before number X triggers an avalanche, and I have seen it myself happening once.

---

## Referee Comment (RC2) · Jürg Schweizer (Referee) · 3 Jul 2019

**Review for TCD**

*Modeling snow slab avalanches caused by weak layer failure – Part II: Coupled mixed-mode criterion for skier triggered anticracks*

by Rosendahl and Weissgraeber

The authors describe a new dry-snow slab avalanche release model. By introducing a Winkler foundation[1] below the slab, the weak layer properties can explicitly be considered (in contrast, for example, to the Heierli model that considers a rigid base below the slab) and the slab displacement field can be described in analytical form.

In part II of their contribution, the authors suggest a coupled mixed-mode criterion of skier triggered anticracks. Whereas classical fracture mechanics assumes a preexisting crack and studies the conditions for crack propagation, the so-called finite fracture mechanics approach, which the authors employ, does not require an initial weakness, but assumes that due to stress overload an initial failure forms provided that sufficient energy is released to form a crack at all. In other words, it is assumed that the stress criterion and the fracture mechanical energy criterion are coupled and need to be satisfied simultaneously.

In general, the authors made an applaudable effort to introduce their model and place it in the context of previous work. I am hesitant in accepting some of their conclusions since they partly reflect some of the assumptions and simplifications inherent to any model. However, if those limitations are properly discussed, I have no principal objections and recommend the paper to be published pending adequate revisions by the authors.

The principles of the model are described in part I. I am not commenting on the first paper – unless reference to it is made in the second part and something is not clear.

1. In the abstract, the authors state that in the limit case of very thick slabs and very steep slopes natural release is obtained. Previous work has shown that natural release cannot be described by a simple stress criterion, even one coupled with a fracture mechanical criterion. Spatial variations in slab and weak layer properties are required to describe natural slab release. Clearly, spatial variations are less decisive for skier-triggering.

2. The concept of finite fracture mechanics assumes that skiers cannot initiate a crack unless sufficient energy is released. Whereas this assumption follows from the model, it is not obvious to me that situations exist where this second conditions is not fulfilled. As far as I understand this means that strength is low, but toughness is high. I am not aware that this scenario is relevant in the case of snow; it seems hypothetical to me.

3. Nevertheless, I agree that in most situations skiers instantaneously induce a macroscopic crack that in most cases is large enough for self-propagation and that no initial weakness is required (as e.g. suggested by Schweizer and Camponovo, 2001) – in contrast to natural release. Natural release is often observed for a load lower than the average stress, which implies that failure starts at locations of below average stability.

4. The authors state on page 7 that the stability of the initial crack is governed by the energy criterion only. Does this mean that the initial finite sized crack $\Delta a$ automatically fulfills the Griffith criterion? If so, I do not understand how Eqs. 8, 19 and the statement on page 11, line 7 relate. Can you please explain why $G_c$ is part of the energy criterion.
* * *
[1] By the way, the Winkler foundation was already introduced by McClung and Borstad (2012) to study avalanche formation.

5. Page 5, line 29: I suggest you introduce a proper reference to part I. In general, I think it is best to make the two contributions self-contained – or merge them.

6. Page 7, line 17: To my understanding, given the continuum mechanical approach, it is best to talk about weak layer failure. Collapse, which I understand as a consequence of failure in a structure that is strong and stiff in compression, but weak in shear, refers to the porous microstructure – not considered in the model.

7. I am not convinced that Eq. 9 represents a suitable strength criterion for the case of snow, a highly anisotropic material.

8. Figure 3: Please more clearly state what kind of experimental data are shown. What means "taken by"? Page 10, lines 1-6: I suggest you consider the strong anisotropy of snow when discussing the failure criteria. I suppose this would change the relative contributions of $G_I$ and $G_{II}$ to $G$.

9. Page 10, line 8-10: both requirements were considered by Gaume and Reuter (2017).

10. Page 12, Table 1: I strongly recommend providing references for the property values presented. For example, the relation between shear, tensile and compressive strength is rather unusual.

11. Page 12, line 7: I suggest using slope angle or incline rather than inclination.

12. Page 12, lines 9-11: It is not clear to me why the critical load in case of the shallow slab on the slope is higher than for the thicker slab. Please explain.

13. Figure 6: As far as I understand the maximum stress at 40 cm depth is always smaller than at 20 cm depth as the additional skier stress decreases with increasing depth. Therefore, I cannot follow the statement that thicker slabs cause larger point loads. Maybe I misunderstand your term critical additional skier load. This should relate to the depth of the weak layer and not to the surface load, since the surface load is a given value, as it is due to a skier. Moreover, I suggest rewording the statement that thicker slabs transfer loads more uniformly. The transfer is always the same.

14. Page 14, line 2 and 6: The stress distribution due to a skier does not depend on the modulus as long as the slab is uniform. Therefore, I cannot follow your argument here. Please explain.

15. Page 18, line 10: To my knowledge, Reuter et al. (2019) did not study the dependence of snow strength properties on strain rate. Some of the studies that explicitly do so include Narita (1983), Schweizer (1998) and Reiweger and Schweizer (2010).

16. Page 18, lines 1-4: I do not understand why the authors state in this context that their approach does not require the assumption of initial flaws in the weak layer; the approach by Gaume and Reuter (2017) does not require this particular assumption either.

17. Page 18, line 10: As mentioned earlier, in my understanding the term collapse refers to the microstructure and is the result of failure; the latter can occur in shear, compression or combined shear and compression. I doubt that one can simply imply from the fact that there is normal deformation, that the failure is compressive.

18. Page 18, lines 13-20: I suggest rewording or partly revisiting this paragraph. For example, the formulation that the critical skier load vanishes in very steep terrain can be misunderstood. As we know by experience triggering is more likely on steeper slopes. In that context, it seems rather counterintuitive that longer cracks are required on steeper slopes.

19. Page 18, line 21: Suggest rewording.

20. Page 18, line 32. Whereas the slab modulus affects the displacement field, the stress due to the skier remains unchanged. Therefore, you have to be careful with using the term bridging that refers to initiation due to skier stress.

21. Page 19, lines 5-15: Whereas I understand that the findings on e.g. crack lengths on slopes results from the model, I doubt whether this specific finding is particular realistic. I suggest revisiting some of the assumptions and discussing them in the light of these results.

22. Page 19, lines 31-24: These section needs to be revisited. If a crack in the flat propagates it is not unusual that a fracture through the slab occurs somewhere. This has frequently been observed. The term shooting crack is used for the situation when cracks propagate. Shooting cracks are best related with avalanche release (Schweizer, 2010).
23. Page 20-21, Conclusions: I suggest you refer also to the limitations of the model and provide an outlook on possible improvements.

References

Gaume, J. and Reuter, B., 2017. Assessing snow instability in skier-triggered snow slab avalanches by combining failure initiation and crack propagation. Cold Reg. Sci. Technol., 144: 6-15.

McClung, D.M. and Borstad, C.P., 2012. Deformation and energy of dry snow slabs prior to fracture propagation. J. Glaciol., 58(209): 553-564.

Narita, H., 1983. An experimental study on tensile fracture of snow. Contrib. Inst. Low Temp. Sci., A32: 1-37.

Reiweger, I. and Schweizer, J., 2010. Failure of a layer of buried surface hoar. Geophys. Res. Lett., 37: L24501.

Reuter, B., Calonne, N. and Adams, E., 2019. Shear failure of weak snow layers in the first hours after burial. The Cryosphere Discussions, 2019: 1-17.

Schweizer, J., 1998. Laboratory experiments on shear failure of snow. Ann. Glaciol., 26: 97-102.

Schweizer, J., 2010. Predicting the avalanche danger level from field observations, Proceedings ISSW 2010. International Snow Science Workshop, Lake Tahoe CA, U.S.A., 17-22 October 2010, pp. 162-165.

Schweizer, J. and Camponovo, C., 2001. The skier's zone of influence in triggering slab avalanches. Ann. Glaciol., 32: 314–320.

3 July 2019/Sz

---

## Author Comment (AC1) · 8 Aug 2019

**Part II: Michael Zaiser**

Prof. Michael Zaiser has raised questions about the concept of FFM and its implications at hand of two thought experiments.

We used the two thought experiments to further explain the concept of FFM – with special focus on size effects.

[Figure]

We thank the referee for the detailed review of our manuscript. We changed manuscript to provide a clearer introduction of FFM and the size effect example.

**Reviewer comments**

The paper deals with the formulation of a failure criterion for collapse-like failure of snow and application to skier triggering of avalanches. It combines a fracture mechanical and a strength-of-materials approach to formulate a criterion for weak layer failure that does not rely on assuming a pre-existing flaw. Fracture mechanical and strength-of-materials ('snow stability index') approaches to snow failure have both been used in the snow literature, and sometimes in a manner that confounds their respective domains of application. Thus a unification of sorts is an inherently desirable undertaking.

The authors first discuss contradictions that may arise from an uncritical application of either strength-of-materials or fracture mechanical criteria to situations where those criteria do not apply. A nice example is given by Eqs. (3) and (4) which are particularly instructive as I have encountered very similarly flawed reasoning in a recent manuscript under review for The Cryosphere: Basically, one cannot simply apply an energy argument to an uncracked specimen and deduce a failure stress from it, nor can one invert the argument and convert a failure stress into a fracture toughness by evaluating the overall energy stored within the sample. By contrast, I find the argument surrounding Figure 1 less convincing. Reference is made to Bazant 1984 but, if we adopt Bazant's reasoning I see no reason why the strength of an uncracked specimen as shown in Figure 1 should be size dependent (in fact, Bazant's relation predicts strength to be size independent at small sample sizes whatever the crack length).

It is important to note that Bazant's [1] arguments are based on the consideration of the microstructure of the material. Hence, Fig. 1 (see Figure 1) in his work does not indicate absolute size on the horizontal axis but a nondimensional size parameter ($\texttt{SIZE} = \lambda = d/d_\mathrm{a}$), which is a characteristic structural dimension $d$ normalized by a

characteristic dimension of the microstructure $d_a$. This is a very important difference to Fig. 1 in our manuscript.

Bazant's Fig. 1 implies that when the size of the structure is of the order of the size of its particles $d/d_a \approx 1$ ($\log d/d_a \approx 0$), failure is governed by stress. When the structure is much larger than its microstructure $d/d_a \gg 1$, the problem is dominated by energy. The structures considered in Fig. 1 in our manuscript fall into the transition zone between the two extremes. Hence, Bazant's work supports our argument that fracture processes are always governed by strength and toughness simultaneously, even if one often hides the other.

In fact, he draws the similar conclusions as FFM. To quote his 1984 paper [1]:

*"2. Dimensional analysis based on the foregoing basic hypothesis shows that, for structures that are geometrically similar (i.e., have the same shape), the nominal stress at failure varies with the structure size as $(1 + \lambda/\lambda_0)^{-1/2}$ where $\lambda_0$ is a constant and $\lambda$ is the ratio of the size of the structure to the maximum size of the aggregate."*

That is, as the size of the structure $d$ decreases, $\lambda$ decreases and, hence, according to the above conclusion, the nominal structural strength increases.

A recent study by Leguillon et al. [2] draws very similar conclusions from FFM. Because the apparent strength of very brittle materials such as ceramics is governed by intrinsic defects such as pores or surface flaws, the authors raised the question of the intrinsic strength of such materials. To answer this, they revisited experimental failure data of surface-flawed metallic ceramics and modeled surface defects of the tested specimens using FFM. The experiments and the FFM model (Figure 2) paint a very similar picture as Bazant's size effect law (Figure 1). That is, when the surface flaws are smaller than the microstructure (obtained, e.g., by surface etching), the problem is governed by (the intrinsic) stress. When the flaws are deep, they act crack-like. The transition resembles

Bazant's size effect law and is well captured by the physical arguments of FFM.

Another example for how FFM describes the transition from stress to energy is given by Weißgraeber et al. [3]. The authors consider ellipses of varying aspect ratios $a/b$ under uniaxial tension (Figure 3). When a narrow ellipsis is oriented perpendicular to the loading direction ($a/b \gg 1$), it behaves crack-like and can be described using linear elastic fracture mechanics (LEFM, dashed line) with the corresponding $1/\sqrt{a}$-size effect. Narrow ellipsis oriented in loading direction ($a/b \ll 1$) represent only very weak stress concentrations and the problem is governed by stress (Strength, dotted line) without size effect. Again the transition ($0.1 \leq a/b \leq 100$) is described by finite fracture mechanics (PM FFM, solid line). The FFM model also captures the size effect of circular holes. This is evident comparing the PM FFM curves at the aspect ratio of $a/b = 1$. Here, the stress at failure is a function of the hole radius $a = b$.

The FFM criterion attempts to resolve the well-known conundrum that exists in theories of fracture, namely that the transition from stress-induced damage accumulation (no crack) to the propagation of a critical crack is not well understood. It does so by combining a strength of materials criterion to obtain an upper estimate of the crack length (a crack can at maximum form over a length where the stress is above sigma_c) with a lower estimate (the ensuing crack must be able to propagate). The crack forms if the lower bound falls below the upper one. Taking the example provided by the authors makes me wonder whether I understand the criterion correctly. Consider the tensile beam of Eqs. (3),(4) of the manuscript. Load this beam at the stress sigma_c such that the stress based criterion for mode I failure is fulfilled over the entire cross section. Finite fracture mechanics seems to indicate that all energy stored in the beam is released if a cross-section spanning crack is formed. If that is true, however, then Eq. (3) entails a critical beam length $l \sim (2EG\_c/sigma\_c^2)$ below which the beam cannot fail because that energy is insufficient. Now I insert typical values of steel, say KIc = 40 MPa m^1/2, sigma_c = 500 MPa, to obtain with EG_c $\sim$ KIc^2 a critical length in the range of 1 cm. Ehem. Do the authors want to imply that my 1cm tensile sample is too

small to break??????? Clearly I must be misunderstanding something.

The suggested thought experiment is an example of a size effect. As shown in many works of Bazant (see, e.g., his review [4]), they exist in virtually any structure (with or without pre-cracks).

Of course also the small tensile steel sample will break. However, it will do so at a slightly higher critical load. A sufficiently long sample will fail at $\sigma = \sigma_{\rm c}$. One that is shorter than the critical length (at which the energy condition is not satisfied anymore), requires a slightly higher load $\sigma > \sigma_{\rm c}$ to fulfill the energy balance for crack nucleation. Hence, as the sample size is further decreased below the critical length, the effective sample strength will increase owing to the energy condition. This argument holds provided the sample is still much larger than its microstructure (so that we find ourselves in the transition zone of Bazant's [1] Fig. 1 size effect law).

I have a second objection. Consider again a thin tensile beam with thickness a and length l ≪ a loaded at a stress slightly above sigma_c. Let us now assume the energy criterion is fulfilled: sigma_c^2 l > 2 E G_c. So the beam breaks. Now consider the same beam but embedded as surface fiber into a bending beam as in Figure 1. Let the bending moment be such that a region of thickness a from the surface is above the critical stress. In that case, the energy release will be less than for the free standing beam, and it may well be that the crack cannot form since sigma_c^2 a < 2 E G_c. However, the stress state in the considered volume is identical in both cases. The only thing which the volume elements (and the microstructure, grains, dislocations, atoms......) in the beam know about the outside world is the local stress acting on them. How do they understand that, in the first case, they should form a crack instantaneously, and in the second case, not?

The fundamental principle of linear elastic fracture mechanics is (global) conservation of energy applied to crack growth. This is reflected in Griffith's crack propagation criterion $\mathcal{G}_{\rm c} = \mathcal{G} = -{\rm d}\Pi/{\rm d}A$, where $\Pi$ is the total potential energy of the system. That

is, Griffith's criterion and, hence, fracture mechanics in general, always considers the global energy balance.

Let us apply your thought experiment to a crack tip: If, as you suggest, volume elements only know local stress acting on them, any crack should grow at arbitrarily small loads because crack tip stresses are singular. However, they do not because the global energy balance (reflected by Griffith's criterion) must be satisfied. Finite fracture mechanics uses the Griffith condition in its original sense where it was defined not only for infinitesimal crack growth and applies it to finite crack extension.

Considering strain energy density locally represents another form of a local criterion (the square of a simple stress criterion). Hence, strain energy density concepts cannot be used to address fracture mechanics problems unless a length scale (a critical distance or an area) is assumed.

I kindly request the authors to clarify the above two points.

Once we accept the basic approach, the development of stress and energy based failure criteria looks sensible. Concerning the skier loading, I have a question since the loading model is not very clearly described. I presume the authors assume plane strain conditions in which case F would need to be a load per unit length. On line 11, page 11 there is a mysterious b which seems not defined.

Yes, we consider the weight load of a skier ($mg$) distributed over an assumed effective length of skis ($l_o$) which provides a force per unit length that corresponds to the load in a unit out-of-plane width plane strain model. If we use an out-of-plane width $b < l_o$ that is not unity, the total force loading of this strip is given by $F = mgb/l_o$. This is explained in part I but was missing in part II. We have addressed this in our revision of part II.

As to material parameter choices, I commented on that point in relation to the companion paper.

In this second part, we only investigate qualitative effects of model parameters. Of

course, the absolute value of model outputs is affected by, e.g., the choice of Young's modulus. However, observations do not change qualitatively. An analysis the model's sensitivity to material parameters, shows, for instance, a rather weak effect of the weak layer's Young's modulus. Hence, it does not seem important to choose a certain set of parameters. We now indicate references for our parameter choices in Table 1.

A minor point: It is noted as an inherent flaw of models that assume subcritical damage accumulation that 'such models would predict avalanche release if only enough skiers ski the same slope in close temporal succession'. While such may not be generic behavior, I cannot see why this should be impossible to happen. I know of several passages in the avalanche literature reporting that several skiers may ski a slope before number X triggers an avalanche, and I have seen it myself happening once.

We agree that this question cannot be resolved conclusively at this point. However, if the general understanding of dry-slab avalanche release comprises damage accumulation by subcritical load, then the consequential release after repeated loading must also be a generally observed feature. The sudden crack formation described by FFM is a though model. It does not necessarily mean that the crack actually appears spontaneously. Instead, by considering only the intact and the fractured state, we can also interpret the crack jump as an accumulation of damage which is just not resolved in time.

[1] Z. P. Bažant. Size Effect in Blunt Fracture: Concrete, Rock, Metal. Journal of Engineering Mechanics, 110(4):518–535, 1984.
[2] D. Leguillon, E. Martin, O. ŠevecÌŇek, and R. Bermejo. What is the tensile strength of a ceramic to be used in numerical models for predicting crack initiation? International Journal of Fracture, 212(1):89–103, 2018.
[3] P. Weißgraeber, J. Felger, D. Geipel, and W. Becker. Cracks at elliptical holes: Stress intensity factor and Finite Fracture Mechanics solution. European Journal of Mechanics - A/Solids, 55:192–198, 2015.
[4] Z. P. Bažant. Size effect on structural strength: a review. Archive of Applied Me-

[Figure]

chanics, 69(9-10):703–725, 1999.

[Figure]

[Figure]

[Figure]

**Fig. 1.** Fig. 1 by Bazant [1]. The x-axis shows a nondimensional size parameter (SIZE = d/d_a), where d is a structural dimension and d_a th e size of the microstructure.

[Figure]

**Fig. 2.** Using FFM, Leguillon et al. [2] show that failure (vertical axies) is governed by stress (horizontal asymptote) when initial flaws (horizontal axis) become smaller than the microstructure.

**Fig. 3.** Stress at failure over aspect ratio a/b of an ellipsis. Transition from stress problem (Strength, dotted) to linear elastic fracture mechanics (LEFM, dashed) captured by FFM (solid)

[Figure]

---

## Author Comment (AC2) · 8 Aug 2019

The reviewer has pointed out several points in the paper where given explanations were insufficient, especially regarding the concept of finite fracture mechanics and the resulting parameter dependences.

We have addressed all points below and improved the manuscript to make the raised points more clear. Since the failure criterion is nonlocal, the results are given in terms of critical (outer) loading on the snow pack. We further detail this and the difference to

a critical local weak layer stress. Further, we now better explain the concept of finite crack lengths and are more specific in the theoretical background of finite cracks and their distinction from critical crack lengths as obtained in, e.g., PSTs.

We thank Dr. Schweizer for his meticulous review. We have addressed all points in detail and changed the manuscript accordingly.

**Reviewer comments**

In general, the authors made an applaudable effort to introduce their model and place it in the context of previous work. I am hesitant in accepting some of their conclusions since they partly reflect some of the assumptions and simplifications inherent to any model. However, if those limitations are properly discussed, I have no principal objections and recommend the paper to be published pending adequate revisions by the authors.

The principles of the model are described in part I. I am not commenting on the first paper – unless reference to it is made in the second part and something is not clear.

1. In the abstract, the authors state that in the limit case of very thick slabs and very steep slopes natural release is obtained. Previous work has shown that natural release cannot be described by a simple stress criterion, even one coupled with a fracture mechanical criterion. Spatial variations in slab and weak layer properties are required to describe natural slab release. Clearly, spatial variations are less decisive for skier-triggering.

When mentioning "natural release" in the abstract, it is our intention to characterize the situation where, according to our model, no additional external (skier) load can be applied under the given conditions. This should not imply that our model provides a prediction of natural release. On the contrary, it is supposed to indicate the limits of the present model. Because this requires some clarification, we removed the statement

from the abstract.

2. The concept of finite fracture mechanics assumes that skiers cannot initiate a crack unless sufficient energy is released. Whereas this assumption follows from the model, it is not obvious to me that situations exist where this second conditions is not fulfilled. As far as I understand this means that strength is low, but toughness is high. I am not aware that this scenario is relevant in the case of snow; it seems hypothetical to me.

Energy release available for crack initiation and crack growth is a structural property. This is evident in the the Griffith criterion $\mathcal{G}_\mathrm{c} = \mathcal{G} = -\mathrm{d}\Pi/\mathrm{d}A$, where $\Pi$ is the total potential energy of the system. That is, the criterion evaluates a global energy balance with respect to crack extension. Hence, toughness on the structural level, i.e., the structural resistance against crack nucleation and crack growth, does not only originate from the material's fracture toughness but also from the amount of energy stored in the structure.

This gives rise to size effects. Small structures store only a small amount of energy so that the energy condition becomes important even when structures without initial flaws are considered. An example is given in Figure 1 in our manuscript. To some extent, the situation resembles a slab that bends owing to skier-loading. The presence of a size effect is evident in the considered experimental data. Finite fracture mechanics explains this size effect without the necessity of assuming large toughness to strength ratios (Weissgraeber et al. [1]).

The comprehensive experimental work of Sigrist [2] reports size effects for both fracture toughness and strength measurements in three-point bending tests of snow beams. This is compelling evidence that both stress and energy are important in the fracture process of snow. The energy balance (first law of thermodynamics) is a fundamental principle. All processes (in our case the transition from the uncracked to the cracked state) must obey this principle.

3. Nevertheless, I agree that in most situations skiers instantaneously induce a macro-
scopic crack that in most cases is large enough for self-propagation and that no initial weakness is required (as e.g. suggested by Schweizer and Camponovo, 2001) – in contrast to natural release. Natural release is often observed for a load lower than the average stress, which implies that failure starts at locations of below average stability.

We agree and do not intend to imply that natural release is covered by our model (see response to point 1). The present model specifically addresses the situation of skier-triggered weak layer failure. We have clarified this in the manuscript.

4. The authors state on page 7 that the stability of the initial crack is governed by the energy criterion only. Does this mean that the initial finite sized crack a automatically fulfills the Griffith criterion? If so, I do not understand how Eqs. 8, 19 and the statement on page 11, line 7 relate. Can you please explain why Gc is part of the energy criterion.

Crack nucleation (no crack $\rightarrow$ finite crack) requires $\overline{\mathcal{G}} \geq \mathcal{G}_c$ (as a necessary but not sufficient condition) (Leguillon [3]). Crack propagation, however, requires $\mathcal{G} = \mathcal{G}_c$ (Broberg [4]). Since the relation between incremental and differential energy release rate is

$$\mathcal{G} = \overline{\mathcal{G}} + a\frac{\partial\overline{\mathcal{G}}}{\partial a}$$

and the present situation is a positive geometry (Weissgraeber et al. [5]), i.e., $\partial\overline{\mathcal{G}}/\partial a > 0$ holds, the differential release rate of initiated cracks will always exceed the fracture toughness. So yes, the initial crack is generally unstable.

However, this is only true in the vicinity of the introduced outer load by the skier. If the crack grows out of the region influenced by skier-loading, the energy release rate decreases (it is now governed by the slab weight only) and the crack can stop (governed by Griffith's criterion). The assessment of the stability of the initiated crack and its ability to propagate in the region that is not affected by the external skier load has to be studied in further research. To do so we could extend our modeling approach with a touchdown condition that eventually leads to an upper bound of the energy release rate.

5. Page 5, line 29: I suggest you introduce a proper reference to part I. In general, I think it is best to make the two contributions self-contained – or merge them.

We have introduced a proper reference to allow for a clear link between the two papers. The two parts are closely interconnected but apart from the definitions of the basic fracture mechanics quantities, they are self-contained.

6. Page 7, line 17: To my understanding, given the continuum mechanical approach, it is best to talk about weak layer failure. Collapse, which I understand as a consequence of failure in a structure that is strong and stiff in compression, but weak in shear, refers to the porous microstructure – not considered in the model.

The denomination "collapse" can be understood differently. Let us develop two thoughts on this topic:

i) Continuum mechanics typically considers boundary-value problems. That is, a problem with a given fixed boundary that cannot change without changing the problem statement. Within the considered boundaries, continuum mechanical constitutive equations describe the material response and do not account for material separation. Hence, they would allow for arbitrarily high loads and do not consider crack nucleation or crack propagation as this creates new boundaries. Yet, we do treat fracture problems using continuum mechanics without the need to consider the microscale at which atomic bonds break (Anderson [6]). This is a very useful approach that can be readily transferred to compressive failure. That is, collapse on the macroscale may be treated just like tensile failure without the consideration of the microstructure.

ii) If we were to consider the microstructure. Collapse can very well be the consequence of pure unidirectional normal loading. Engineering structure such as rods, beams, plates and shells can buckle. They lose structural stability and exhibit a sudden sideways deflection at a critical (pure) compressive load (Gross et al. [7]). For a brittle and low-strength material like snow (ice), buckling can be expected to be accompanied by structural failure. Of course, superimposed shear loading is likely to facilitate

structural failure. However, collapse does not require shear.

The strong anisotropy of weak layers concerning normal and shear strength is of course a direct consequence of the porous microstructure . This can be readily accounted for in continuum mechanical models.

7. I am not convinced that Eq. 9 represents a suitable strength criterion for the case of snow, a highly anisotropic material.

This is a reasonable remark. We are not aware of a conclusive study about mixed-mode strength criteria for weak layers and it is not the scope of the present work to investigate the topic. It is just our intention do demonstrate that FFM works with different mixed-mode strength hypotheses.

8. Figure 3: Please more clearly state what kind of experimental data are shown. What means "taken by"? Page 10, lines 1-6: I suggest you consider the strong anisotropy of snow when discussing the failure criteria. I suppose this would change the relative contributions of GI and GII to G.

We have expanded the description of the experimental data shown in Figure 3 and payed tribute to the strong anisotropy of snow in the discussion of the failure criteria.

9. Page 10, line 8-10: both requirements were considered by Gaume and Reuter (2017).

Their work makes use of an empirical equation obtained from a fit to numerical analyses to round rigid particles with interaction criterion. This length is considered to be the critical length of a crack required for unstable propagation. The second length is the size of the overloaded area below a concentrated skier load. The strength condition and the stability of the crack are considered in an uncoupled fashion.

Since they do so, we must ask the question: How does their model explain size effects? And what happens if a crack is short enough so that I should not propagate according to the energy criterion but according to the strength criterion it will?

[Figure]

These are questions that FFM provides a definitive physical answer to. It provides a unique solution based on experimentally measurable material properties. It is not possible to solve one equation for the length scale and another one for the critical load because both unknowns appear in both equations which, hence, are implicitly coupled.

10. Page 12, Table 1: I strongly recommend providing references for the property values presented. For example, the relation between shear, tensile and compressive strength is rather unusual.

We have provided references for the exemplary values chosen. In the Figures 6, 7 and 9 we show and discuss the general parameter dependences.

11. Page 12, line 7: I suggest using slope angle or incline rather than inclination.

We have changed the wording as suggested.

12. Page 12, lines 9-11: It is not clear to me why the critical load in case of the shallow slab on the slope is higher than for the thicker slab. Please explain.

Considering remark 13 below, it seems there is a misunderstanding of our denomination "critical additional skier load". As depicted in the pictograph (sketch in the top right corner of, e.g., Figure 5) and indicated by the unit (kilo-newtons), the y-axis label "Critical additional skier load $F$ [kN]" in our diagrams refers to the surface point load (concentrated force load) that, when applied, triggers weak layer failure.

A weak layer below a slab is loaded by the slab's weight which causes stresses in the weak layer. An additional concentrated force load applied to the slab's surface (e.g., by a skier) increases the weak layer stresses. The "critical additional skier load $F$" denotes the critical value of this additional surface load at which our model predicts anticrack nucleation within the weak layer. The term "point load" does not refer to a "maximum stress" at a certain point within the weak layer, as you assume in remark 13. It denotes a concentrated force.

In other words, if a concentrated force $F$ that is smaller than the value indicated in

our diagrams, is applied on the surface the snowpack, the entire snowpack would stay intact. With reference to remark 13, the force load $F$ a skier needs to apply to the snowpack to trigger failure is not a given value but computed as the result of the present model.

In order to avoid this misunderstanding for the readers, we changed the denomination to "critical skier force $F$" in the manuscript added the following paragraph to the beginning of the results section:

> *In each study slabs loaded by their own weight and an additional concentrated force are considered. The force represents the outer load that a skier imposes on the snowpack. The given failure criterion predicts the critical magnitude of this additional concentrated force that leads to anticrack nucleation in the weak layer. We call this failure initiating force the critical skier force.*

Concerning remark 12, the critical skier force $F$, i.e., the load bearing capacity of the snowpack is higher in flat terrain because the weak layer normal strength and mode I fracture toughness are higher than the corresponding material properties in shear. We discussed this on page 18, lines 13-20 in out manuscript:

> The lower strength of the weak layer in shear governs the decrease of the critical loading with slope angle.

13. Figure 6: As far as I understand the maximum stress at 40 cm depth is always smaller than at 20 cm depth as the additional skier stress decreases with increasing depth. Therefore, I cannot follow the statement that thicker slabs cause larger point loads. Maybe I misunderstand your term critical additional skier load. This should relate to the depth of the weak layer and not to the surface load, since the surface load

is a given value, as it is due to a skier. Moreover, I suggest rewording the statement
that thicker slabs transfer loads more uniformely. The transfer is always the same.

See our response to remark 12 for a clarification of the term "critical additional skier
load".

We combine a local stress criterion with a global energy balance of fracture mechanics.
The given coupled stress and energy criterion in the framework of finite fracture me-
chanics is all in all a global criterion that cannot be evaluated locally. So the resulting
quantity must be a global quantity as well. On the y-axis, we show the critical outer
(surface) load onto the snowpack that leads to failure of the weak layer. The transfer
of this load from the point where it introduced by the skier through the snowpack to
the weak layer changes with the bending stiffness of the slab. This is an effect of the
layering. Because the stiff slab rests on a compliant weak layer, its bending stiffness
governs the overall snowpack deformation and, hence, weak layer stresses.

We have further clarified this in the manuscript and changed the wording.

14. Page 14, line 2 and 6: The stress distribution due to a skier does not depend on
the modulus as long as the slab is uniform. Therefore, I cannot follow your argument
here. Please explain.

The restriction "as long as the slab is uniform" is important, must hold in thickness
direction and actually extends through the weak layer. If we consider a homogeneous
elastic half space, stresses within this homogeneous body that originate from force
loads, are indeed independent of the Young's modulus. However, a stiff slab on a soft
weak layer is by no means a homogeneous body. Further, we are not interested in
stresses within the homogeneous slab but in stresses within the soft weak layer.

When the Young's modulus of the weak layer is smaller than the slab's Young's modu-
lus, the load transfer depends on the Young's modulus of the slab. By load transfer we
refer to the resulting weak layer stresses owing to a surface load.

Think of a stiff steel plate and a rather soft plastic board (slabs of different Young's moduli) resting on a mattress (weak layer), say both plates 1×1 m and 2 cm thick and the mattress 2×2 m and 10 cm thick. When an 80 kg person steps onto the center of the steel plate, its deformation will be hardly visible. Stresses within the mattress are low because the weight of 80 kg is transferred rather homogeneously over the entire 1 m$^2$. However, if the same person steps onto the plastic board, it will deform (bend) considerably causing localized stress below the person. This effect is reflected in Figure 7.

We have elaborated this in the manuscript.

15. Page 18, line 10: To my knowledge, Reuter et al. (2019) did not study the dependence of snow strength properties on strain rate. Some of the studies that explicitly do so include Narita (1983), Schweizer (1998) and Reiweger and Schweizer (2010).

In the work by Reuter et al. [8] strain rate effects have been studied to some extent (cf. Figure 7 of their work). However, we agree that the works of Narita [9], and Reiweger and Schweizer [10] should be referenced at that point, since they provide insight into the strain rate dependence of the failure mechanisms.

16. Page 18, lines 1-4: I do not understand why the authors state in this context that their approach does not require the assumption of initial flaws in the weak layer; the approach by Gaume and Reuter (2017) does not require this particular assumption either.

We state this in this context because it is an important property of robust models. However, we agree that it sounds like we imply that Gaume and Reuter [11] did make assumptions about initial flaws. We have rephrased the paragraph accordingly. However, concerning the model by Gaume and Reuter [11], it shall be pointed at the concerns raised in response to remark 9.

[Figure]

17. Page 18, line 10: As mentioned earlier, in my understanding the term collapse

refers to the microstructure and is the result of failure; the latter can occur in shear, compression or combined shear and compression. I doubt that one can simply imply from the fact that there is normal deformation, that the failure is compressive.

In a mechanical perspective, buckling denotes a primary macroscopic stability failure mechanism owing to pure normal loading. It is directly and inseparably linked to an abrupt (compressive) deformation of the structure. Hence, when pure normal deformation is present, pure compressive failure is possible and likely.

However, there is no need to distinguish different microscopic failure modes on the continuum level. We agree that microstructural failure may occur in shear, compression or combined shear and compression. Yet, the aim of continuum mechanical models is to smear microstructural effects over the macroscopic scale. This is done by considering different compressive and shear strengths as well as different mode I and mode II fracture toughnesses. Hence, on the continuum level, we denote macroscopic failure owing to macroscopic normal deformation compressive.

18. Page 18, lines 13-20: I suggest rewording or partly revisiting this paragraph. For example, the formulation that the critical skier load vanishes in very steep terrain can be misunderstood. As we know by experience triggering is more likely on steeper slopes. In that context, it seems rather counterintuitive that longer cracks are required on steeper slopes.

This remark relates to the previously discussed misunderstanding of our denomination "critical skier load $F$". As we pointed out in our response to remark 12, the term "critical additional skier load" refers to the outer force load that a snowpack can carry on top of its own weight before an anticrack nucleates in the weak layer. Hence, a vanishing critical skier load in steep terrain corresponds to the described observation: Triggering is more likely on steeper slopes because the "critical additional skier load" required to initiate weak layer cracks is lower (and vanishes on very steep slopes). In other words, on steep slopes a very small load can trigger weak layer failure.

We agree that longer cracks in steeper slopes are somewhat counterintuitive. However, it is to note that the length of nucleated finite cracks provides no information on how likely or unlikely anticracks can be triggered. Only the critical skier load, which is also a result of the model, does. Both load and crack length are results of the complex interaction between stress and energy within the finite fracture mechanics failure model. In the present case, the crack length becomes long because the mode II energy release rate (shear) is much smaller than the mode I energy release rate (collapse). There is a number of factors that may cause this behavior: i) We did not consider weak layer anisotropy, ii) we considered normal and shear deformation uncoupled and iii) we estimated the mode II fracture toughness $\mathcal{G}_{\mathrm{IIc}}$. Once all these points are addressed in a refined future model (which we are already working on), we may obtain a different picture.

We have improved the paragraph clarifying the denomination "critical skier load" and discussing the crack lengths.

19. Page 18, line 21: Suggest rewording.

Please refer to our answer to remark 14. Deformations of a homogeneous slab on a soft weak layer depend on the slab's bending stiffness $EI$. This stiffness has a linear dependence on the slab's Young's modulus $E$ and a cubic dependence on its height $I = bh^3/12$ (assuming a rectangular cross-section). The thought experiment suggested in response to remark 14 can be done with two plates of different thickness, as well. A 0.1 mm thick steel plate on a soft mattress will deform when an 80 kg person steps onto it, a 10 cm steel plate will not. We have, therefore, revised the paragraph to better explain our understanding of load transfer.

20. Page 18, line 32. Whereas the slab modulus affects the displacement field, the stress due to the skier remains unchanged. Therefore, you have to be careful with using the term bridging that refers to initiation due to skier stress

As discussed in response to point 14, the Young's modulus of the (stiff) slab does affect

the stresses in the (soft) weak layer. The (bending) deformations of the slab change and hence the strains and stresses in the weak layer, too.

21. Page 19, lines 5-15: Whereas I understand that the findings on e.g. crack lengths on slopes results from the model, I doubt whether this specific finding is particular realistic. I suggest revisiting some of the assumptions and discussing them in the light of these results.

The statement that the shear fracture toughness $\mathcal{G}_{\text{IIc}}$ is much smaller than the compressive fracture toughness $\mathcal{G}_{\text{Ic}}$ agrees with experimental findings on other materials such as glassy foams (Heierli [12]). The specific identified ratio of $\mathcal{G}_{\text{Ic}}/\mathcal{G}_{\text{IIc}} \approx 20 \mathrel{..} 40$ likely depends on assumptions of the present work, e.g., isotropy of the weak layer and uncoupled normal and shear deformations. Unfortunately, no experimental data is available to unambiguously determine the ratio and to test the validity of our assumptions. Therefore, we just added a discussion of effects of model assumptions to the paragraph, as suggested.

22. Page 19, lines 31-24: These section needs to be revisited. If a crack in the flat propagates it is not unusual that a fracture through the slab occurs somewhere. This has frequently been observed. The term shooting crack is used for the situation when cracks propagate. Shooting cracks are best related with avalanche release (Schweizer, 2010).

We have removed the confusing statement.

Page 20-21, Conclusions: I suggest you refer also to the limitations of the model and provide an outlook on possible improvements.

We have added the following limitations to our conclusions:

- Homogeneous slab

- Uncoupled normal and shear displacements

- Isotropic weak layer

As an outlook the following improvements are vital:

- Layered slab

- Coupled normal and shear displacements

- Measurements of elastic snow properties, in particular weak layer tensile and shear moduli

- Measurements of failure and fracture envelopes

[1] P. Weißgraeber, D. Leguillon, and W. Becker. A review of Finite Fracture Mechanics: crack initiation at singular and non-singular stress raisers. Archive of Applied Mechanics, 86(1-2):375–401, 2016.

[2] C. Sigrist. Measurement of fracture mechanical properties of snow and application to dry snow slab avalanche release. PhD thesis, ETH Zürich, 2006.

[3] D. Leguillon. Strength or toughness? A criterion for crack onset at a notch. European Journal of Mechanics – A/Solids, 21(1):61–72, 2002.

[4] K. B. Broberg. Cracks and fracture. Elsevier, 1999.

[5] P. Weißgraeber, S. Hell, and W. Becker. Crack nucleation in negative geometries. Engineering Fracture Mechanics, 168:93–104, 2016.

[6] T. L. Anderson. Fracture Mechanics. CRC Press, Boca Raton, 4th edition, 2017.

[7] D. Gross, W. Hauger, J. Schröder, and W. A. Wall. Technische Mechanik 2. Springer-Lehrbuch. Springer Berlin Heidelberg, Berlin, Heidelberg, 2014.

[8] B. Reuter, N. Calonne, and E. Adams. Shear failure of weak snow layers in the first hours after burial. The Cryosphere Discussions, (January):1–17, 2019.

[9] H. Narita. An experimental study on tensile fracture of snow. Contributions from the Institute of Low Temperature Science, Series A, 32:1–37, 1983.

[10] I. Reiweger and J. Schweizer. Failure of a layer of buried surface hoar. Geophysical Research Letters, 37(24):L24501, 2010.

[11] J. Gaume and B. Reuter. Assessing snow instability in skier-triggered snow slab avalanches by combining failure initiation and crack propagation. Cold Regions Science and Technology, 144(May):6–15, 2017.

[12] J. Heierli, P. Gumbsch, and D. Sherman. Anticrack-type fracture in brittle foam under compressive stress. Scripta Materialia, 67(1):96–99, 2012.

---

## Referee Report (RR1)

**Review for TCD**

*Modeling snow slab avalanches caused by weak layer failure – Part II: Coupled mixed-mode criterion for skier triggered anticracks (revised version)*

by Rosendahl and Weissgraeber

I am in general pleased with the replies, but I think some of the replies could have been better reflected in the manuscript. I suggest the authors consider the following points:

P1, L12:    I repeat that "collapse" is a poor term and it is not consistently used in the manuscript. Collapse is a generic term for the breakdown of a structure and does not refer to a specific failure mode. I suggest using failure, e.g. here on page 1, line 12, and compression, e.g. when referring to mode II (compression) vs. mode I (shear). Collapse is an essential process in relation to weak layer failure under mixed-mode loading and explains crack propagation in low-angle terrain.
Of course, it is up to the authors to decide on how to use the term. Adding a definition might help easing potential confusion.

Table 1:     I suggested adding references to the choice of material parameters in Table 1.
I acknowledge that the authors followed this advice. However, some of the references are simply other numerical studies that used similar values. I would expect, e.g. for fracture toughness, that the authors actually refer to corresponding experimental work.

P12, L6-9:   In the reply to my previous question 4, the authors states that the initial crack is generally unstable, and that the differential energy release rate always exceeds the fracture toughness. On page 12, lines 6-9 the authors write that the size of the initial crack is different from the critical crack size. I still think some clarification here would improve the manuscript.

P12, L11:    Please provide a definition and value for $b$, which I guess is the out-of-plane (ski) width. Also, it would be helpful to provide a typical value of $F$ so that the results on critical skier force (e.g. in Figure 5) can be put into context.
Also, for clarification I suggest that at least once you refer to the surface, e.g., on page 12, line 11: … for static skier loading the local force acting onto the snow surface F = ….

P12, L20:    "… where the out load will … " please consider rewording to improve clarity.

P20, L9-12:  By the way, slab fractures were also addressed by Gaume et al. (2015) and Reuter and Schweizer (2018).

Davos, 3 November 2019
Jürg Schweizer.

---

## Author Response (AR2)

**Response to additional remarks by Jürg Schweizer**
**Review report Nov 03 2019**

1.: I repeat that "collapse" is a poor term and it is not consistently used in the manuscript. Collapse is a generic term for the breakdown of a structure and does not refer to a specific failure mode. I suggest using failure, e.g. here on page 1, line 12, and compression, e.g. when referring to mode II (compression) vs. mode I (shear). Collapse is an essential process in relation to weak layer failure under mixed-mode loading and explains crack propagation in low-angle terrain. Of course, it is up to the authors to decide on how to use the term. Adding a definition might help easing potential confusion.

We agree that further explanation of the definition of the term collapse increases clarity. In the sense of anticrack fracture mechanics the term collapse describes the compaction of the weak layer due to the collapse of its microstructure. This allows for the free deformation of the crack faces and it can be triggered not only by compressive loading. As in classical fracture mechanics, anticracks can also be driven by pure mode II or as in the present case by mixed-mode. The microstructural damage processes might differ but a compaction can also occur. We added the following clarification to the introduction:

> Here, collapse refers to the sudden loss of volume of the porous weak layer, which can be caused by pure compression or mixed-mode shear and compression loading.

2.: Table 1: I suggested adding references to the choice of material parameters in Table 1. I acknowledge that the authors followed this advice. However, some of the references are simply other numerical studies that used similar values. I would expect, e.g. for fracture toughness, that the authors actually refer to corresponding experimental work.

In particular concerning geometric properties the choice of dimensions is rather arbitrary within physically reasonable domains. Hence, we chose other numerical studies as references to allow for direct comparison of practical implications of our model and other theoretical/numerical works.

Fracture toughness cannot be measured directly and always relies on precise models that use experimentally accessible quantities (as e.g. a load at failure or the critical cut length) to provide the energy release rate as an output. Hence, we complemented the model reference with the reference to the experimental data used.

3.: In the reply to my previous question 4, the authors states that the initial crack is generally unstable, and that the differential energy release rate always exceeds the fracture toughness. On page 12, lines 6-9 the authors write that the size of the initial crack is different from the critical crack size. I still think some clarification here would improve the manuscript.

The initiated finite crack is generally unstable since for the present case the incremental energy release is always higher than the differential energy release rate. And since the incremental energy release rate must exceed the fracture toughness to allow for the initiation of finite cracks (cf. coupled criterion eq. (8), the differential energy release rate will fulfill the Griffith criterion for crack growth ($\mathcal{G} \geq \mathcal{G}_c$). But with further distance to the skier force onto the slab the energy release rate decreases and crack arrest can occur. The stability of the cracks and their propagation is a wider topic to be addressed in subsequent work.

We have extended this discussion in the manuscript in section 3.

> The size of the initial crack is different from the critical crack length $a_c$ as used, e.g., by Gaume (2017). $\Delta a$ does not represent the critical crack length for crack propagation. It is the size of initial weak layer collapse owing to overcritical skier loading. The stability of this crack must be analyzed in a second step.

> The second of the two necessary conditions for crack nucleation that are coupled in the coupled

criterion (8) is the Griffith energy criterion for *finite* cracks (5). That is, when crack nucleation is predicted this condition is always fulfilled. The stability of the initiated crack must be assessed using the Griffith criterion for *infinitesimal* growth of an existing crack, $\mathcal{G} = \mathcal{G}_c$ (Broberg, 1999). The relation between incremental and differential energy release rates is given by

$$\mathcal{G} = \overline{\mathcal{G}} + a \frac{\partial \overline{\mathcal{G}}}{\partial a}. \tag{1}$$

Further, the present situation is a locally-positive geometry (Weissgraeber et al., 2016a; Sapora and Cornetti, 2018). That is, the energy release rate increases with crack length in the vicinity of the skier, i.e., $\partial \overline{\mathcal{G}}/\partial a > 0$. Hence, on account of Eq. (1), $\mathcal{G} > \mathcal{G}_c$ holds true and the nucleated crack $\Delta a$ is initially unstable. Of course, the complete analysis of the stability of initiated cracks must also include further aspect such as the touchdown condition of the slab onto the collapsed weak layer, which limits the energy release outside the skier-influenced zone, and must be studied in future work.

4.: Please provide a definition and value for b, which I guess is the out-of-plane (ski) width. Also, it would be helpful to provide a typical value of F so that the results on critical skier force (e.g. in Figure 5) can be put into context. Also, for clarification I suggest that at least once you refer to the surface, e.g., on page 12, line 11: ... for static skier loading the local force acting onto the snow surface F = ....

You are right, for the evelution of the critical skier force the model width is set to the effective out-of-plane ski length. We also added the reference to the snow surface, typical values of $F$ and the definition of $b$ as the out-of-plane model width, as suggested. The updated sentence now reads:

For static skier loading the local load acting on the snow surface is $F = mgb/l_o$ where $l_o$ is the effective out-of-plane length of the object, such as the length of skis, and $b$ is the out-of-plane width of the model. In the present work we use $b = l_o$ to investigate the total force applied by a skier, which yields $F \approx 0.8\,\text{kN}$ for an 80 kg skier using $b = l_o = 1\,\text{m}$ (Table 1).

5.: "... where the out load will ... " please consider rewording to improve clarity.

We corrected the typo. The corrected sentence now reads:

... where the out*er* load will lead to ...

6.: By the way, slab fractures were also addressed by Gaume et al. (2015) and Reuter and Schweizer (2018).

We have changed the reference.

[revised manuscript text omitted]